# Dynamic and Chemical Constraints to Enhance the Molecular Masked Graph Autoencoders

**Jiahui Zhang[1,2], Wenjie Du[1,2,†], Yang Wang[1,2,†]**
[1]University of Science and Technology of China, China
[2]Suzhou Institute for Advanced Research, USTC, China
kongping@mail.ustc.edu.cn   {duwenjie, angyan}@ustc.edu.cn

## Abstract

Masked Graph Autoencoders (MGAEs) have gained significant attention recently. Their proxy tasks typically involve random corruption of input graphs followed by reconstruction. However, in the molecular domain, two main issues arise: the predetermined mask ratio and reconstruction objectives can lead to suboptimal performance or negative transfer due to overly simplified or complex tasks, and these tasks may deviate from chemical priors. To tackle these challenges, we propose Dynamic and Chemical Constraints (DyCC) for MGAEs. This includes a masking strategy called GIBMS, which preserves essential semantic information during graph masking while adaptively adjusting the mask ratio and content for each molecule. Additionally, we introduce a Soft Label Generator (SLG) that reconstructs masked tokens as learnable prototypes (soft labels) rather than hard labels. These components adhere to chemical constraints and allow dynamic variation of proxy tasks during training. We integrate the model-agnostic DyCC into various MGAEs and conduct comprehensive experiments, demonstrating significant performance improvements. Our code is available at `https://github.com/forever-ly/DyCC`.

## 1 Introduction

Molecular Representation Learning (MRL) plays a pivotal role in many related applications such as drug discovery, material design, and reaction prediction [8, 11, 44]. By representing molecules as graphs, where atoms are treated as nodes and bonds as edges, Graph Neural Networks (GNNs) [43, 20, 14] have exhibited remarkable performance across a wide range of tasks. However, a significant challenge is the scarcity of labeled data, which limits the effectiveness of supervised learning. Inspired by the remarkable progress in self-supervised pretraining in natural language processing [12], Masked Graph Autoencoders (MGAEs) [17, 41, 25, 16] have arisen as a promising approach to addressing these challenges. The pioneering work [17] on this topic introduced the pretraining of GNNs using a mask-then-reconstruction task called AttrMask. Specifically, they randomly mask some proportions of atoms and then pretrain the models to predict them. AttrMask has emerged as a fundamental pretraining task and many subsequent works [16, 54] adopt it as a subtask for pretraining. Despite their success, we have identified two limitations that still lack exploration.

**The first limitation is that proxy tasks are predetermined and lack dynamic adaptability.** The effectiveness of MGAEs is largely governed by their proxy tasks, which are defined by the graph-masking strategy and the reconstruction objective. Prior approaches rely on fixed mask strategies and tokenizers, which result in suboptimal performance. **From the perspective of input corruption**, the

---

† corresponding author

39th Conference on Neural Information Processing Systems (NeurIPS 2025).

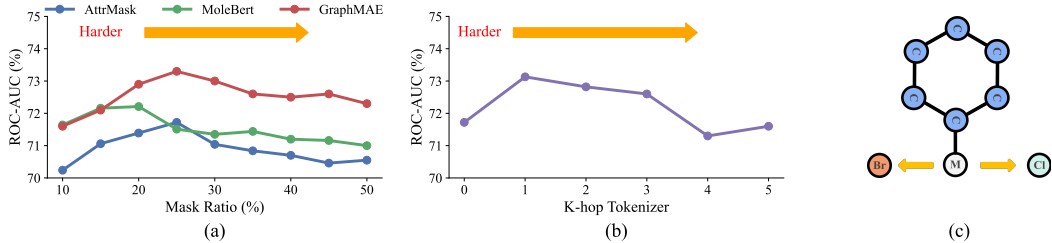

Figure 1: **(a)** The difficulty of proxy tasks for the AttrMask, MoleBert, and GraphMAE models was increased by raising the mask ratio. The average ROC-AUC scores were computed across the eight classification datasets in MoleculeNet [39]. **(b)** The average ROC-AUC scores of AttrMask with $K$-hop subgraph token as the reconstruction target. The larger the value of $K$, the more challenging the proxy task becomes. **(c)** Reconstructing the masked atom as either bromine or chlorine can generate valid molecules.

masking strategy significantly impacts pretraining performance. As shown in Fig. 1(a), involving GraphMAE[16], AttrMask [17], and MoleBert [41], we progressively increased the mask ratio to make the proxy task harder. Initially, as the mask ratio increased, pretraining performance improved; however, beyond a certain threshold, performance deteriorated. **From the perspective of the reconstruction target**, the choice of reconstruction tokens greatly affects pretraining performance. AttrMask treats atoms as tokens, but the simplicity of its reconstruction objective can cause suboptimal performance or even negative transfer. Consequently, several studies have proposed more challenging tokenizers [54, 41, 25]. For example, SimSGT [25] uses $K$-hop subgraphs as reconstruction targets and employs a simple GNN-based tokenizer to generate subgraph-level tokens. By adjusting the value of $K$, the complexity of the reconstruction task can be controlled—the larger the $K$, the more difficult the task. However, we observe that more challenging tokenizers do not necessarily lead to better pretraining performance. As shown in Fig. 1(b), we used $K$-hop subgraph tokens as the reconstruction target (with $K = 0$ corresponding to node-level tokens). While increasing $K$ indicates a higher tokenizer complexity (i.e., a harder proxy task), the performance did not consistently improve. **In summary**, employing fixed masking and reconstruction strategies (i.e., a predetermined tokenizer) necessitates a cumbersome search for optimal hyperparameters—and even then yields only suboptimal results. A more effective solution is to introduce dynamic adaptability into proxy tasks.

**The second limitation is that proxy tasks may not adhere to the constraints imposed by chemical priors**, potentially leading to nonsensical or even harmful self-supervised learning signals. We outline three aspects that deviate from chemical constraints (CC), labeled as *CC1, CC2,* and *CC3*. *CC1: Each molecule has its specific mask ratio*. Currently, most approaches adopt a globally fixed mask ratio. However, different molecules exhibit varying degrees of structural redundancy, rendering a uniform mask ratio suboptimal—too high for certain molecules. *CC2: The importance of atoms within a molecule varies*, as certain key atoms play pivotal roles in determining molecular functional properties, reactivity, and biological interactions. Consequently, prioritizing the masking of these important atoms can encourage the model to engage in more challenging contextual reasoning, thereby fostering a deeper understanding of critical structural features. *CC3: The reconstruction target should not be unique*. Given the vast chemical compound space, many compounds exhibit significant structural similarity. As illustrated in Fig. 1(c), the masked atom could be reconstructed as either Cl or Br. However, current methods often constrain the reconstruction by favoring specific atom types (or substructures), producing conflicting self-supervised signals.

To address these issues, we propose Dynamic and Chemical Constraints (DyCC) MGAEs, a model-agnostic approach dynamically adjusts the proxy task while adhering to chemical constraints. Specifically, we leverage the Graph Information Bottleneck (GIB) theory [52, 51] to redefine graph masking as a graph compression problem, introducing the GIB-based Mask Strategy (GIBMS). The core concept is to identify a compressed core substructure within the molecular graph that encapsulates its key properties, and to place greater emphasis on reconstructing these core substructures during the graph masking stage. This design explicitly addresses *CC1* and *CC2* by encouraging the model to focus on essential structural information while learning robust molecular representations. However, traditional GIB relies on supervisory signals, which are unavailable during pretraining when downstream task

labels are not accessible. **Our contribution lies in extending GIB theory to the unsupervised setting**. We adopt the common multi-view assumption [30, 42], enabling us to demonstrate that the mutual information between the molecular graph and self-supervised signals acts as a lower bound for the mutual information between the graph and downstream task labels. As a result, we reformulate the problem as maximizing the mutual information between the molecular graph and self-supervised signals, which can be achieved through a contrastive learning paradigm.

Additionally, we introduce the Soft Label Generator (SLG) module, **which transforms the reconstruction objective from a specific token (hard label) to a soft cluster assignment (soft label), thereby fulfilling *CC3*[7, 4, 2].** We define potential clusters as prototypes represented by learnable vectors and subsequently map the hard labels to probability distributions of these prototypes using the Sinkhorn-Knopp algorithm[10]. Specifically, we randomly initialize a set of prototypes (learnable vectors), and both the token labels and the reconstruction predictions are evaluated for similarity against these prototypes, yielding two probability assignment matrices (soft labels). Minimizing the discrepancy between these two matrices is equivalent to minimizing the difference between the labels and the reconstruction predictions. During the training process, as the prototypes are updated, the mapped soft labels dynamically change, thereby enabling the proxy task to be adjusted throughout the training phase (fulfilling *dynamic*). In extreme cases, when the assignment probability distributions converge to the one-hot distribution, the soft labels degenerate into hard labels, resulting in high inter-class distinctiveness and simplifying the reconstruction task. Conversely, as the assignment probability distributions approach a uniform distribution, inter-class separability diminishes, rendering the reconstruction task exceptionally challenging.

In summary, our core contributions are as follows: First, we identified the lack of dynamism and the failure to adhere to chemical constraints in the proxy tasks of MGAEs. To address these challenges, we extend the supervised GIB theory to the unsupervised setting and design the GIBMS module, generating the optimal mask ratio and mask content for each molecule. Additionally, we introduce the concept of soft assignment into the graph reconstruction stages, avoiding conflicting self-supervised signals and dynamically adjusting the tokenizer. Lastly, we integrate these two model-agnostic modules into multiple MGAEs. Extensive experimentation shows consistent improvements across the integrated models, validating the effectiveness and generality of our approach.

## 2   Related Work

**Graph Contrastive Self-supervised Learning**   Contrastive self-supervised learning, follows the principle of mutual information maximization [1], which typically works to maximize the correspondence between the representations of an instance (e.g., node, subgraph, or graph) in its different augmentation views. GraphCL [49] performs graph-level contrastive learning with combinations of four graph augmentations, namely node dropping, edge perturbation, subgraph cropping, and feature masking. InfoGraph [32] conducts graph representation learning by maximizing the mutual information between graph-level representations and local substructures. GraphLOG [33] leverages clustering to construct hierarchical prototypes of graph samples. They further contrast each local instance with its corresponding higher prototype for contrastive learning. JOAO [48] proposes a framework to automatically search proper data augmentations for GCL. GraphMVP [24] uses a contrastive loss and a generative loss to connect the 2-dimensional view and 3-dimensional view of the same molecule, in order to inject the 3-dimensional knowledge into the 2-dimensional graph encoder. RGCL [22] trains a rationale generator to identify the causal subgraph in graph augmentation. Although the contrastive learning paradigm is very successful, it relies on data augmentation, which depends on domain knowledge.

**Graph Generative self-supervised learning**   Generative self-supervised learning focuses on recovering missing parts of input data. It can be further divided into two families: autoregressive and autoencoding models. Autoregressive models break down joint probability distributions into a product of conditionals. In supervised graph generation, earlier methods like GraphRNN [47] and GCPN [46] have been proposed. More recently, GPT-GNN [18] represents an attempt to incorporate graph generation as a training objective. On the other hand, graph autoencoders are designed to reconstruct input data without enforcing a decoding order. Early work in this field includes GAE and VGAE [21], which use 2-layer GNN as encoders and dot-product decoding for link prediction. AttrMask [17] adopts a random masking strategy, where a portion of the nodes are randomly masked,

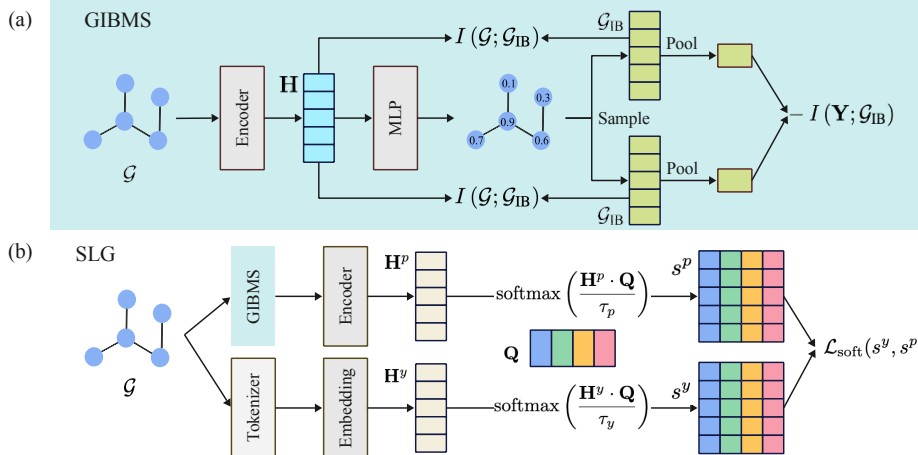

Figure 2: Overall architecture of DyCC. **(a):** The GIBMS module identifies the sampling probabilities of nodes in the molecular graph, based on graph information bottleneck theory. **(b):** During reconstruction, both the hard labels of the tokens $\mathbf{H}^y$ and the predictions $\mathbf{H}^p$ are mapped into soft labels $s^y$ and soft predictions $s^p$ through the SLG module, forming probability distributions. The training objective $\mathcal{L}_{\text{soft}}$ is to minimize the distance between these two distributions.

and the goal is to reconstruct them. GraphMAE [16] propose to focus on feature reconstruction with both a masking strategy and scaled cosine error that benefit the robust training of GraphMAE. Mole-BERT [41] observes that mask atom prediction is an overly easy pretraining task. Therefore, they employ a GNN tokenizer pretrained by VQ-VAE to generate more complex reconstruction targets for masked atom modeling. SimSGT [25] utilizes a simple GNN-based tokenizer, which removes the nonlinear update function in each GNN layer to derive subgraph level tokens. These works all adopt a fixed proxy task setup and do not take into account the three chemical constraints we proposed.

## 3 Method

To meet the constraints of chemical priors and enable dynamic adaptability in proxy tasks, we have designed the GIBMS and SLG, as depicted in Fig. 2. Next, we will provide detailed explanations of these two modules.

### 3.1 GIBMS: Graph Information Bottleneck for Mask Strategy

Subgraph recognition [52, 51, 53] aims to identify a condensed core substructure within a graph that maximizes its informativeness regarding the graph property while discarding redundant information. Inspired by this, we propose to dynamically generate a core subgraph for each molecule (CC1) and place greater emphasis on masking the important atoms within this core substructure (CC2). Subgraph recognition can be formulated by optimizing GIB [52] with a mutual information estimator. For a graph $\mathcal{G}$ and its label information $\mathbf{Y}$, the optimal IB-graph $\mathcal{G}_{\text{IB}}$ which keeps minimal sufficient information:

$$\mathcal{G}_{\text{IB}} = \arg\min_{\mathcal{G}_{\text{IB}}} - I\left(\mathbf{Y}; \mathcal{G}_{\text{IB}}\right) + \beta I\left(\mathcal{G}; \mathcal{G}_{\text{IB}}\right) \tag{1}$$

where $\beta$ serves as a Lagrangian multiplier to balance the two mutual information terms.

**Obtaining node representations of molecular graph**   To begin, it is essential to convert the molecular graph $\mathcal{G}$ into a vector representation. GNNs have emerged as the predominant approach for handling molecular graphs due to their effectiveness in capturing graph-structured data. Let $\Phi$ denote a GNN encoder, which is used to generate the node-level representations of the graph. Formally, this process can be expressed as:

$$\mathbf{H} = \Phi(\mathcal{G}) \tag{2}$$

where $\mathbf{H}$ represents the node representations of the graph.

**Injecting Noise to Obtain the IB-Graph** The discrete nature of graphs renders direct acquisition of the $\mathcal{G}_{\text{IB}}$ impractical due to the exponential proliferation of candidates ($2^N$) for a given graph $\mathcal{G}$ with $N$ nodes. Therefore, we propose a relaxation by assuming a gilbert random graph [13], where node selection from the input graph $\mathcal{G}$ is conditionally independent. This assumption enables us to factorize the probability of the $\mathcal{G}_{\text{IB}}$ as:

$$\mathcal{P}(\mathcal{G}_{\text{IB}}|\mathcal{G}) = \prod_{i \in \mathcal{V}} \mathcal{P}(\mathcal{V}_i|\mathcal{G}) \tag{3}$$

Here, $\mathcal{V}$ denotes the nodes of $\mathcal{G}$, and $\mathcal{P}(\mathcal{V}_i|\mathcal{G})$ signifies the probability distribution for node $\mathcal{V}_i$. A straightforward instantiation of $\mathcal{P}(\mathcal{V}_i|\mathcal{G})$ is the Bernoulli distribution $\mathcal{V}_i \sim \text{Bernoulli}(p_i)$ parameterized by $p_i$. We compute the probability $p$ for each node based on its embedding $\mathbf{H}$ using a Multi-Layer Perceptron network $\mathcal{M}$, expressed as:

$$p = \text{Sigmoid}(\mathcal{M}(\mathbf{H})) \tag{4}$$

Here, the Sigmoid function ensures the probabilities are normalized. The graph $\mathcal{G}_{\text{IB}}$ is then derived by performing Bernoulli sampling on all nodes:

$$\mathcal{G}_{\text{IB}} = \{\mathcal{V}_i \mid \mathcal{V}_i \sim \text{Bernoulli}(p_i), i = 1, 2, \ldots, N\} \tag{5}$$

Here $\mathcal{G}_{\text{IB}}$ is a sampled combination of nodes. Furthermore, we adopt noise injection following VGIB [51] to optimize the subgraph, as it has been proven to mitigate inefficiency and instability in GIB optimization caused by mutual information estimation. The key idea is to introduce more noise into less informative subgraphs while injecting less noise into more informative ones. Specifically, with the calculated probability $p$, we perturb the representation $\mathbf{H}$ by adding noise $\epsilon$:

$$\hat{\mathbf{H}} = \lambda \mathbf{H} + (1 - \lambda) \epsilon \tag{6}$$

where $\lambda \sim \text{Bernoulli}(p)$ and $\epsilon \sim \mathcal{N}(\mu_{\mathbf{H}}, \sigma_{\mathbf{H}}^2)$. Here, $\mu_{\mathbf{H}}$ and $\sigma_{\mathbf{H}}^2$ are the mean and variance of $\mathbf{H}$, respectively, and Bernoulli represents the Bernoulli distribution. Thus, the information of $\mathcal{G}$ is compressed into $\mathcal{G}_{\text{IB}}$ with the probability of $\lambda$ by replacing non-important nodes with noise. Moreover, to make the sampling process differentiable, we adopt a gumbel-sigmoid [26, 19] for discrete random variable $\lambda$, i.e.,

$$\lambda = \text{Sigmoid}(\frac{1}{t} \log\left[\frac{p}{(1-p)}\right] + \log\left[\frac{u}{(1-u)}\right]) \tag{7}$$

where $u \sim \text{Uniform}(0, 1)$, and $t$ is the temperature hyperparameter. To ensure that the obtained $\mathcal{G}_{\text{IB}}$ is meaningful, we need to solve Eq. (1). Next, we will discuss the optimization of the first prediction term $-I(\mathbf{Y}; \mathcal{G}_{\text{IB}})$ and the second compression term $I(\mathcal{G}; \mathcal{G}_{\text{IB}})$ separately.

**Minimizing the Prediction Term** The first term, $-I(\mathbf{Y}; \mathcal{G}_{\text{IB}})$, encourages $\mathcal{G}_{\text{IB}}$ to be informative about the label $\mathbf{Y}$. Since the goal of pre-training is to enhance the performance of downstream tasks, $\mathbf{Y}$ here refers to the labels of the downstream task dataset. To avoid confusion in notation moving forward, we will denote the labels of the downstream tasks as $\mathbf{Y}^{\text{sup}}$. Given the input $\mathcal{G}_{\text{IB}}$ and the downstream task labels $\mathbf{Y}^{\text{sup}}$, our objective is to learn a vector representation of $\mathcal{G}_{\text{IB}}$, denoted as $\mathbf{Z}_{\mathcal{G}_{\text{IB}}}^{\text{sup}} = \text{Pool}(\hat{\mathbf{H}})$, which can effectively predict the labels $\mathbf{Y}^{\text{sup}}$. Here, Pool represents the pooling function.

$$\mathbf{Z}_{\mathcal{G}_{\text{IB}}}^{\text{sup}} = \arg\max_{\mathbf{Z}_{\mathcal{G}_{\text{IB}}}} I(\mathbf{Z}_{\mathcal{G}_{\text{IB}}}; \mathbf{Y}^{\text{sup}}) \tag{8}$$

However, since GIBMS is applied during the pretraining stage, obtaining downstream task labels $\mathbf{Y}^{\text{sup}}$ is not feasible. Here we approach it from a self-supervised learning perspective. Considering the input $\mathcal{G}_{\text{IB}}$ and self-supervised signals $\mathbf{S}$ (e.g., augmentations of $\mathcal{G}_{\text{IB}}$) as two different views of the data, we aim to derive sufficient self-supervised representations $\mathbf{Z}_{\mathcal{G}_{\text{IB}}}^{\text{ssl}}$ that can maximize the preservation of shared information between the views.

$$\mathbf{Z}_{\mathcal{G}_{\text{IB}}}^{\text{ssl}} = \arg\max_{\mathbf{Z}_{\mathcal{G}_{\text{IB}}}} I(\mathbf{Z}_{\mathcal{G}_{\text{IB}}}; \mathbf{S}) \tag{9}$$

where $\mathbf{Z}_{\mathcal{G}_{\text{IB}}}$ is the representation of the graph, obtained by pooling the node representations $\hat{\mathbf{H}}$, i.e., $\mathbf{Z}_{\mathcal{G}_{\text{IB}}} = \text{Pool}(\hat{\mathbf{H}})$, where Pool is the pooling function. By adopting the common multi-view assumption [30, 42], we have (Appendix E.1):

$$I(\mathcal{G}_{\text{IB}}; \mathbf{Y}^{\text{sup}}) = I(\mathbf{Z}_{\mathcal{G}_{\text{IB}}}^{\text{sup}}; \mathbf{Y}^{\text{sup}}) \geq I(\mathbf{Z}_{\mathcal{G}_{\text{IB}}}^{\text{ssl}}; \mathbf{Y}^{\text{sup}}) \geq I(\mathcal{G}_{\text{IB}}; \mathbf{Y}^{\text{sup}}) - \epsilon_{\text{info}}; \quad \epsilon_{\text{info}} > 0 \tag{10}$$

In this paper, we assume that with appropriate self-supervised signals $\mathbf{S}$, $\epsilon_{\text{info}}$ is negligible. Consequently, the above formula suggests that the self-supervised learned representations $\mathbf{Z}^{\text{ssl}}_{\mathcal{G}_{\text{IB}}}$ can capture almost as much task-relevant information about $\mathbf{Y}^{\text{sup}}$ as the supervised representations $\mathbf{Z}^{\text{sup}}_{\mathcal{G}_{\text{IB}}}$. In this case, minimizing $-I(\mathcal{G}_{\text{IB}}; \mathbf{Y}^{\text{sup}})$ is approximately equivalent to maximizing $I(\mathbf{Z}^{\text{sup}}_{\mathcal{G}_{\text{IB}}}; \mathbf{S})$. Since recently proposed contrastive learning methods [35, 15, 50], which aim to pull positive samples closer and push negative samples apart in the representation space, have been theoretically proven to maximize the mutual information between positive pairs, we leverage this approach. Given $\mathcal{G}_{\text{IB},i}$, we can repeatedly sample from Eq. (5) to obtain its positive sample $\mathcal{G}'_{\text{IB},i}$ as the self-supervised signal $\mathbf{S}$. Then, we can maximize $I(\mathbf{Z}^{\text{sup}}_{\mathcal{G}_{\text{IB}}}; \mathbf{S})$ using a contrastive learning loss, such as InfoNCE [37]:

$$I\left(\mathbf{Y}^{\text{sup}}; \mathcal{G}_{\text{IB}}\right) = \mathcal{L}_{\text{pred}}\left(\mathbf{Y}^{\text{sup}}, \mathcal{G}_{\text{IB}}\right) = -\frac{1}{K}\sum_{i=1}^{K}\log\frac{\exp\left(\sin\left(\mathbf{Z}_{\mathcal{G}_{\text{IB},i}}, \mathbf{Z}_{\mathcal{G}'_{\text{IB},i}}\right)/\tau\right)}{\sum_{j=1,j\neq i}^{K}\exp\left(\sin\left(\mathbf{Z}_{\mathcal{G}_{\text{IB},i}}, \mathbf{Z}_{\mathcal{G}_{\text{IB},j}}\right)/\tau\right)} \quad (11)$$

Here, the representations $\mathbf{Z}_{\mathcal{G}_{\text{IB},i}}$ and $\mathbf{Z}_{\mathcal{G}'_{\text{IB},i}}$ of two IB-graphs $\mathcal{G}_{\text{IB},i}$ and $\mathcal{G}'_{\text{IB},i}$ are considered as positive samples, while representations $\mathbf{Z}_{\mathcal{G}_{\text{IB},j}}$ of $\mathcal{G}_{\text{IB},j}$ from other graphs in the same batch are treated as negative samples. $K$ and $\tau$ indicate the number of paired graphs in a batch and the temperature hyperparameter, respectively.

**Optimizing the Compression Term** The second term minimizes the mutual information of $\mathcal{G}$ and $\mathcal{G}_{\text{IB}}$ so that $\mathcal{G}_{\text{IB}}$ only receives limited information from the input graph $\mathcal{G}$. We can derive its variational upper bound (see Appendix E.2):

$$I\left(\mathcal{G}; \mathcal{G}_{\text{IB}}\right) \leq \mathbb{E}_{\mathcal{G}}\left(-\frac{1}{2}\log A + \frac{1}{2N}A + \frac{1}{2N}B^2\right) = \mathcal{L}_{\text{comp}}\left(\mathcal{G}, \mathcal{G}_{\text{IB}}\right) \quad (12)$$

where $N$ represents the number of nodes in $\mathcal{G}$, $A = \sum_{j=1}^{N}\left(1 - \lambda_j\right)^2$ and $B = \frac{\sum_{j=1}^{N}\lambda_j(\mathbf{H}_j - \mu_{\mathbf{H}})}{\sigma_{\mathbf{H}}}$.

**The Final Training Objective** Based on the aforementioned analysis, the training objective of GIBMS is as follows:

$$\mathcal{L}_{\text{total}} = \mathcal{L}_{\text{pred}}\left(\mathbf{Y}^m, \mathcal{G}_{\text{IB}}\right) + \mathcal{L}_{\text{pred}}\left(\mathcal{G}_{\text{IB}}, \mathbf{Y}^{\text{sup}}\right) + \beta\mathcal{L}_{\text{comp}}\left(\mathcal{G}, \mathcal{G}_{\text{IB}}\right) \quad (13)$$

**Dynamic Masking Strategy** The trained GIBMS is able to generate an importance score $p_i$ for each atom node $v_i$ in the molecular graph $\mathcal{G}$ according to Eq.(7), where $p_i$ represents the probability of retaining the node under the information bottleneck framework. Subsequently, we adopt the Gumbel-Sigmoid approximation to sample a dynamic masking factor $\lambda_i$ for each node, as shown in Eq.(7). Here, $\lambda_i$ is a continuous variable in the range $[0, 1]$, which is further converted into a binary masking decision by applying a threshold $r$, where $\lambda_i > r$ indicates that the corresponding atom will be masked. This strategy allows the masking ratio to be adaptively adjusted at the molecule level according to the specific structural characteristics of each graph, without the need to predefine a fixed global masking ratio (CC1). Moreover, the mechanism tends to preferentially mask nodes with higher importance scores (CC2), thereby encouraging the pretraining model to focus on capturing the key structures within molecular graphs and enhancing the discriminability and generalization of the learned representations.

### 3.2 Soft Label Generator

The output of the tokenizer can be either discrete values (e.g., atomic numbers) or continuous vectors (e.g., representations of subgraphs). For the sake of convenience in subsequent discussions, we map the categorical variable $y$ of tokens through embedding into a vector, unifying the reconstruction label as a $d$-dimensional vector $\mathbf{H}^y \in \mathbb{R}^d$.

Regardless of the tokenizer used, existing methods reconstruct masked atoms into specific types. In the preceding discussion, we emphasized the importance of soft labels. The question now arises: *how do we obtain soft label $s^y$?* Since we cannot access all molecules in the world, it is not feasible to directly acquire the probability of reconstructing a masked atom into different tokens. To address

this limitation, we introduce soft label assignment [7, 4, 2]. We assume the existence of $n$ learnable latent prototypes $\mathbf{Q} \in \mathbb{R}^{n \times d}$, where each masked atom has a probability of being reconstructed into all these prototypes. Firstly, we map hard labels $\mathbf{H}^y$ to soft labels $s^y$:

$$s^y = \mathrm{softmax}\left(\frac{\mathbf{H}^y \cdot \mathbf{Q}}{\tau_y}\right) \tag{14}$$

where $\tau_y \in (0, 1)$ is a temperature. Similarly, for the node representations $\mathbf{H}^p$ outputted by the encoder $\Phi'$, we generate their predictions $s^p$ by measuring the cosine similarity to the same prototype matrix $\mathbf{Q}$ with temperature $\tau_p \in (0, 1)$.

$$s^p = \mathrm{softmax}\left(\frac{\mathbf{H}^p \cdot \mathbf{Q}}{\tau_p}\right) \tag{15}$$

Note, we always choose $\tau_y < \tau_p$ to encourage sharper target predictions, which implicitly guides the model to produce confident low entropy anchor predictions. We penalize when the prediction $s^p$ is different from the soft label $s^y$. We enforce this criterion using a standard cross-entropy loss $H\left(s^y, s^p\right)$. We also incorporate the mean entropy maximization (ME-MAX) regularizer [3, 5] to encourage the model to utilize the full set of prototypes. Denote the average prediction of a batch of $M$ samples as $\bar{s^p}$ :

$$\bar{s^p} = \frac{1}{M} \sum_{i=1}^{K} s_i^p \tag{16}$$

The ME-MAX regularizer simply seeks to maximize the entropy of $\bar{s^p}$, denoted $H(\bar{s^p})$, or equivalently, minimize the negative entropy of $\bar{s^p}$. Thus, the overall objective is:

$$\mathcal{L}_{\mathrm{soft}}(s^y, s^p) = \frac{1}{K} \sum_{i=1}^{K} H\left(s_i^y, s_i^p\right) - \alpha H(\bar{s^p}) \tag{17}$$

where $\alpha > 0$ controls the weight of the ME-MAX regularization.

## 4 Experiments

### 4.1 Experiments setup

**Pretraining setup**  For the pretraining stage, we utilized 2 million molecules sourced from the ZINC15 database [31], following the precedent of prior studies [17]. The GIBMS module was trained using the loss function defined in Eq. (13), where the temperature factor $\tau = 0.1$ for the InfoNCE loss, and $\beta = 0.01$ controls the trade-off between prediction and compression. After training the GIBMS module, we utilized it to generate corresponding mask probabilities for each atom of the 2 million molecules in ZINC15 and sampled masked atoms based on these probabilities. In the reconstruction phase, we mapped the hard labels outputted by the tokenizer to soft labels using the SLG module. By default, we set temptures $\tau_p = 0.25$, $\tau_y = 0.1$, and the number of prototypes $n = 128$. After pretraining, we employed the widely-adopted 8 binary classification datasets within MoleculeNet [39] to evaluate performance on downstream molecular property prediction tasks (see Appendix B). These downstream datasets are divided into train/valid/test sets using scaffold split by 8:1:1 to facilitate an out-of-distribution evaluation setting. We report the mean performances (ROC-AUC) and standard deviations on the downstream datasets across ten random seeds.

**Baselines**  We integrated our method into three MGAEs: AttrMask [17], MoleBert [41], and SimSGT [25]. All our settings remain consistent with the configurations of these models. It is noteworthy that, for a fairer evaluation of DyCC on the masked atom modeling proxy task, we have excluded irrelevant enhancements of MoleBert and SimSGT. Specifically, we removed the GraphTrans variant from SimSGT to ensure consistency in using the GIN architecture. Additionally, we eliminated the triplet masked contrastive learning (TMCL) from MoleBert as it is unrelated to MGAEs (see Appendix C.5). Furthermore, we selected several other self-supervised graph pretraining models for further comparison, including InfoGraph [32], GPT-GNN [18], EdgePred [17], ContextPred [17], GraphLOG [33], G-Contextual [29], G-Motif [29], AD-GCL [34], JOAO [48], SimGRACE [40], GraphCL [49], GraphMAE [16], GraphMVP [24] and MGSSL [54]. The results are collect from MoleBert [41].

Table 1: Transfer learning ROC-AUC (%) scores on eight MoleculeNet datasets. The suffix "DyCC" implies the introduction of both the GIBMS and SLG modules.

| Dataset | Tox21 | ToxCast | Sider | ClinTox | MUV | HIV | BBBP | Bace | Avg.($\uparrow$) |
|---|---|---|---|---|---|---|---|---|---|
| No Pretrain | $74.6_{\pm0.4}$ | $61.7_{\pm0.5}$ | $58.2_{\pm1.7}$ | $58.4_{\pm6.4}$ | $70.7_{\pm1.8}$ | $75.5_{\pm0.8}$ | $65.7_{\pm3.3}$ | $72.4_{\pm3.8}$ | 67.0 |
| InfoGraph | $73.3_{\pm0.6}$ | $61.8_{\pm0.4}$ | $58.7_{\pm0.6}$ | $75.4_{\pm4.3}$ | $74.4_{\pm1.8}$ | $74.2_{\pm0.9}$ | $68.7_{\pm0.6}$ | $74.3_{\pm2.6}$ | 70.1 |
| GPT-GNN | $74.9_{\pm0.3}$ | $62.5_{\pm0.4}$ | $58.1_{\pm0.3}$ | $58.3_{\pm5.2}$ | $75.9_{\pm2.3}$ | $65.2_{\pm2.1}$ | $64.5_{\pm1.4}$ | $77.9_{\pm3.2}$ | 68.5 |
| EdgePred | $76.0_{\pm0.6}$ | $64.1_{\pm0.6}$ | $60.4_{\pm0.7}$ | $64.1_{\pm3.7}$ | $75.1_{\pm1.2}$ | $76.3_{\pm1.0}$ | $67.3_{\pm2.4}$ | $77.3_{\pm3.5}$ | 70.1 |
| ContextPred | $73.6_{\pm0.3}$ | $62.6_{\pm0.6}$ | $59.7_{\pm1.8}$ | $74.0_{\pm3.4}$ | $72.5_{\pm1.5}$ | $75.6_{\pm1.0}$ | $70.6_{\pm1.5}$ | $78.8_{\pm1.2}$ | 70.1 |
| GraphLoG | $75.0_{\pm0.6}$ | $63.4_{\pm0.6}$ | $59.6_{\pm1.9}$ | $75.7_{\pm2.4}$ | $75.5_{\pm1.6}$ | $76.1_{\pm0.8}$ | $68.7_{\pm1.6}$ | $78.6_{\pm1.0}$ | 71.6 |
| G-Contextual | $75.0_{\pm0.6}$ | $62.8_{\pm0.7}$ | $58.7_{\pm1.0}$ | $60.6_{\pm5.2}$ | $72.1_{\pm0.7}$ | $76.3_{\pm1.5}$ | $69.9_{\pm2.1}$ | $79.3_{\pm1.1}$ | 69.3 |
| G-Motif | $73.6_{\pm0.7}$ | $62.3_{\pm0.6}$ | $61.0_{\pm1.5}$ | $77.7_{\pm2.7}$ | $73.0_{\pm1.8}$ | $73.8_{\pm1.1}$ | $66.9_{\pm3.1}$ | $73.0_{\pm3.3}$ | 70.2 |
| AD-GCL | $74.9_{\pm0.4}$ | $63.4_{\pm0.7}$ | $61.5_{\pm0.9}$ | $77.2_{\pm2.7}$ | $76.3_{\pm1.4}$ | $76.7_{\pm1.2}$ | $70.7_{\pm0.3}$ | $76.6_{\pm1.5}$ | 72.2 |
| JOAO | $74.8_{\pm0.6}$ | $62.8_{\pm0.7}$ | $60.4_{\pm1.5}$ | $66.6_{\pm3.1}$ | $76.6_{\pm1.7}$ | $76.9_{\pm0.7}$ | $66.4_{\pm1.0}$ | $73.2_{\pm1.6}$ | 69.7 |
| SimGRACE | $74.4_{\pm0.3}$ | $62.6_{\pm0.7}$ | $60.2_{\pm0.9}$ | $75.5_{\pm2.0}$ | $75.4_{\pm1.3}$ | $75.0_{\pm0.6}$ | $71.2_{\pm1.1}$ | $74.9_{\pm2.0}$ | 71.2 |
| GraphCL | $75.1_{\pm0.7}$ | $63.0_{\pm0.4}$ | $59.8_{\pm1.3}$ | $77.5_{\pm3.8}$ | $76.4_{\pm0.4}$ | $75.1_{\pm0.7}$ | $67.8_{\pm2.4}$ | $74.6_{\pm2.1}$ | 71.2 |
| GraphMAE | $75.2_{\pm0.9}$ | $63.6_{\pm0.3}$ | $60.5_{\pm1.2}$ | $76.5_{\pm3.0}$ | $76.4_{\pm2.0}$ | $76.8_{\pm0.6}$ | $71.2_{\pm1.0}$ | $78.2_{\pm1.5}$ | 72.3 |
| GraphMVP | $74.9_{\pm0.8}$ | $63.1_{\pm0.2}$ | $60.2_{\pm1.1}$ | $79.1_{\pm2.8}$ | $77.0_{\pm0.6}$ | $76.0_{\pm0.1}$ | $70.8_{\pm0.5}$ | $79.3_{\pm1.5}$ | 72.6 |
| MGSSL | $75.2_{\pm0.6}$ | $63.3_{\pm0.5}$ | $61.6_{\pm1.0}$ | $77.1_{\pm4.5}$ | $77.6_{\pm0.4}$ | $75.8_{\pm0.4}$ | $68.8_{\pm0.6}$ | $78.8_{\pm0.9}$ | 72.3 |
| AttrMask | $75.1_{\pm0.9}$ | $63.3_{\pm0.6}$ | $60.5_{\pm0.9}$ | $73.5_{\pm4.3}$ | $75.8_{\pm1.0}$ | $75.3_{\pm1.5}$ | $65.2_{\pm1.4}$ | $77.8_{\pm1.8}$ | 70.8 |
| AttrMask-DyCC | $\mathbf{76.6}_{\pm0.5}$ | $\mathbf{64.6}_{\pm0.4}$ | $\mathbf{61.3}_{\pm0.6}$ | $\mathbf{79.8}_{\pm3.5}$ | $\mathbf{76.7}_{\pm0.9}$ | $\mathbf{77.6}_{\pm1.2}$ | $\mathbf{70.5}_{\pm1.0}$ | $\mathbf{82.1}_{\pm2.0}$ | **73.7** |
| Mole-BERT | $76.2_{\pm0.5}$ | $63.9_{\pm0.3}$ | $61.4_{\pm1.9}$ | $75.1_{\pm3.0}$ | $77.4_{\pm2.1}$ | $77.5_{\pm1.0}$ | $66.8_{\pm1.5}$ | $78.9_{\pm0.9}$ | 72.2 |
| Mole-BERT-DyCC | $\mathbf{76.3}_{\pm0.5}$ | $\mathbf{64.4}_{\pm0.5}$ | $\mathbf{61.4}_{\pm0.9}$ | $\mathbf{78.9}_{\pm2.4}$ | $\mathbf{78.6}_{\pm1.9}$ | $\mathbf{77.7}_{\pm0.9}$ | $\mathbf{70.8}_{\pm0.6}$ | $\mathbf{82.2}_{\pm0.9}$ | **73.8** |
| SimSGT | $75.1_{\pm0.5}$ | $63.5_{\pm0.4}$ | $61.0_{\pm0.4}$ | $79.1_{\pm2.6}$ | $76.0_{\pm0.5}$ | $76.3_{\pm0.5}$ | $70.9_{\pm0.6}$ | $82.5_{\pm0.9}$ | 73.0 |
| SimSGT-DyCC | $\mathbf{76.0}_{\pm0.5}$ | $\mathbf{64.6}_{\pm0.4}$ | $\mathbf{61.6}_{\pm0.6}$ | $\mathbf{80.5}_{\pm2.2}$ | $\mathbf{77.7}_{\pm0.9}$ | $\mathbf{77.3}_{\pm0.8}$ | $\mathbf{71.4}_{\pm0.7}$ | $\mathbf{83.4}_{\pm1.0}$ | **74.1** |

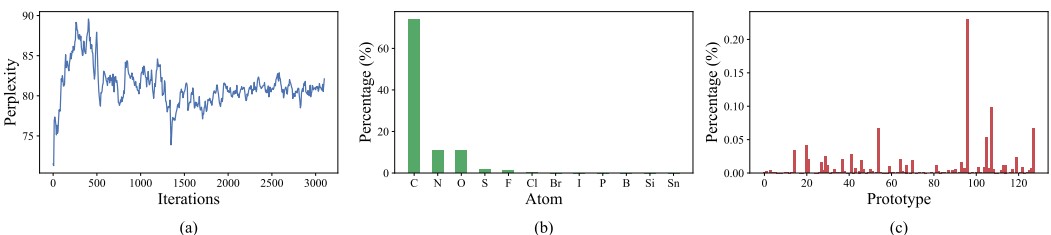

Figure 3: **(a)**: The variation of perplexity of prototypes during the training process. **(b)**: The atoms ratios of various chemical elements in the ZINC datasets. **(c)**: The distribution of 128 prototypes in the ZINC datasets after training.

## 4.2 Model performance

As depicted in Table 1, the incorporation of DyCC into AttrMask, MoleBert, and SimSGT significantly boosts the performance of pretraining. Particularly noteworthy is its integration into SimSGT, where it surpasses the "No pretrain" model by 10.4%, achieving a new state-of-the-art result. Moreover, DyCC demonstrates effective mitigation of the impact of tokenizers on pretraining. Previously, substantial performance variations were observed among the original AttrMask, MoleBert, and SimSGT models due to differences in tokenizers. However, following the integration of DyCC, these performance gaps narrow considerably, indicating reduced reliance on tokenizers. This can be attributed to DyCC's adaptive adjustment of the reconstruction targets, consequently diminishing dependency on tokenizers. In Appendix C.4, we verified the efficacy of DyCC across a broader spectrum of downstream tasks and datasets, including four molecular property prediction regression tasks and two Drug-Target Affinity (DTA) regression tasks [27, 28].

## 4.3 Detailed Analysis of GIBMS and SLG

Here, we primarily analyze the role of GIBMS and SLG, while experiments on other hyperparameters can be found in Appendix C.6.

**The role of GIBMS module.** To further evaluate GIBMS, we employed the MUTAG dataset, which includes 4,337 molecular graphs, each classified into one of two categories based on its mutagenic effect. As noted in GNNExplainer [45], carbon rings with chemical groups $NH_2$ or $NO_2$ are known to be mutagenic. We labeled the top 30% most important atoms identified by GIBMS, and if these

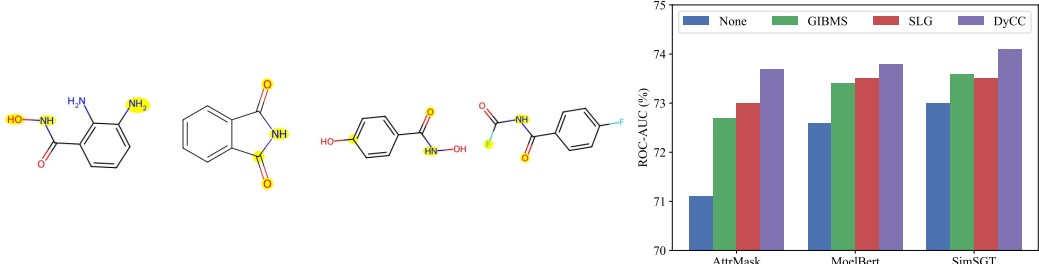

Figure 4: (Left) Core substructures (highlighted) extracted by GIBMS for four molecules. (Right) Component ablation of GIBMS and SLG.

atoms included $NH_2$ or $NO_2$, it was deemed a success. The success rate of GIBMS was 74%, demonstrating the effectiveness of the GIBMS module. In addition, we conducted a qualitative analysis of substructures based on our prior chemical knowledge. As depicted in Fig. 4, we randomly selected four molecules and utilized GIBMS to extract core substructures. The results indicate that the model tends not to focus on aromatic rings but rather tends to discover the substructures around them. This finding aligns with chemical knowledge, as aromatic rings, which contribute to the stability of molecules, are not directly related to chemical properties, whereas substructures in the side chains are more likely to contain chemical information.

**The role of SLG module.** SLG is proposed for dynamically adjusting task difficulty. To investigate its effectiveness, we utilize perplexity as an evaluation metric to assess the probability of different prototypes being utilized. A higher perplexity suggests a more uniform utilization of prototypes, implying increased difficulty in the reconstruction task. As depicted in Fig. 3(a), perplexity initially increases during training, then gradually decreases and converges to a stable value. This indicates that SLG enables our model to dynamically adjust the difficulty of the reconstruction task. Moreover, SLG effectively mitigates issues such as small vocabulary size and token imbalance. For instance, in the widely used ZINC15 dataset, which comprises 12 types of atoms, with 95% of the atoms distributed among the top three atom types (Fig. 3(b)), SLG allows flexible specification of the number of prototypes (determined by $n$), and yields a more uniform distribution of tokens, as illustrated in Fig. 3(c).

**Are both GIBMS and SLG necessary.** To investigate this, we separately added only one module, either GIBMS or SLG, into AttrMask, MoleBert, and SimSGT. As depicted in Fig. 4, we observed that while introducing either module alone improves the effectiveness of pretraining across all MGAEs, combining both strategies leads to better results.

# 5   Conclusion

We identified two significant issues when applying existing MGAEs methods to the molecular domain. On one hand, the proxy tasks are predetermined and lack the capability for dynamic adjustment during training. On the other hand, there are designs that do not align with chemical priors. To address these challenges, we propose the DyCC framework, which consists of two modules: GIBMS and SLG. The GIBMS module employs graph information bottleneck theory to identify nodes that preserve semantics during masking, enabling adaptive masking. The SLG module utilizes a set of learnable prototypes to map the hard labels of tokens to soft labels, dynamically updating these soft labels throughout the training process. This allows the reconstruction objectives to adaptively adjust as well. We integrated DyCC into various existing MGAEs, significantly enhancing pre-training performance while reducing reliance on tokenizers.

# 6   Acknowledgement

This paper is partially supported by the National Natural Science Foundation of China (No.12227901). The AI-driven experiments, simulations and model training were performed on the robotic AI-Scientist

platform of Chinese Academy of Sciences., Anhui Science Foundation for Distinguished Young Scholars (No.1908085J24), Natural Science Foundation of China (No.62502491).

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

# A  Limitations

When training the GIBMS module, we adopt the common multi-view assumption [30, 42], assuming that our self-supervised proxy task is sufficiently effective to yield a small $\epsilon_{info}$. However, this assumption may not always hold. Despite this, we empirically observe that leveraging this assumption indeed benefits the training of GIBMS. Therefore, we proceed with this assumption.

# B  Details of molecular datasets

We provide detailed information of the datasets for molecular property prediction (classification and regression) and drug target affinity prediction in Table 2.

Table 2: Summary for the molecule datasets for downstream tasks.

| Dataset | Task | # Tasks | # Molecules | # Proteins | # Molecule-Protein |
|---|---|---|---|---|---|
| BBBP | Classification | 1 | 2,039 | – | – |
| Tox21 | Classification | 12 | 7,831 | – | – |
| ToxCast | Classification | 617 | 8,576 | – | – |
| Sider | Classification | 27 | 1,427 | – | – |
| ClinTox | Classification | 2 | 1,478 | – | – |
| MUV | Classification | 17 | 93,087 | – | – |
| HIV | Classification | 1 | 41,127 | – | – |
| Bace | Classification | 1 | 1,513 | – | – |
| Delaney | Regression | 1 | 1,128 | – | – |
| Lipo | Regression | 1 | 4,200 | – | – |
| Malaria | Regression | 1 | 9,999 | – | – |
| CEP | Regression | 1 | 29,978 | – | – |
| Davis | Regression | 1 | 68 | 379 | 30056 |
| KIBA | Regression | 1 | 2,068 | 229 | 118,254 |

**Molecule representations.** For simplicity, we use a minimal set of node and bond features that unambiguously describe the two-dimensional structure of molecules following previous works. We use RDKit o obtain these features, as show in Table 3 and Table 4.

Table 3: Atom features.

| features | size | description |
|---|---|---|
| atom type | 100 | type of atom (e.g., C, N, O), by atomic number |
| formal charge | 5 | integer electronic charge assigned to atom |
| number of bonds | 6 | number of bonds the atom is involved in |
| chirality | 5 | number of bonded hydrogen atoms |
| number of H | 5 | number of bonded hydrogen atoms |
| atomic mass | 1 | mass of the atom, divided by 100 |
| aromaticity | 1 | whether this atom is part of an aromatic system |
| hybridization | 5 | sp, sp2, sp3, sp3d, or sp3d2 |

Table 4: Bond features.

| features | size | description |
|---|---|---|
| bond type | 4 | single, double, triple, or aromatic |
| stereo | 6 | none, any, E/Z or cis/trans |
| in ring | 1 | whether the bond is part of a ring |
| conjugated | 1 | whether the bond is conjugated |

**Dataset Splitting.** We apply the scaffold splitting for all tasks on all datasets. It splits the molecules with distinct two-dimensional structural frameworks into different subsets. It is a more challenging but practical setting since the test molecular can be structurally different from training set. Here we apply the scaffold splitting to construct the train/validation/test sets.

## C    Experimental Details

### C.1    Computational resources

Our experiments are conducted using an NVIDIA DGX A100 server. Each experiment can be executed on a single GPU while staying within the limit of 30 GB of GPU memory consumption.

### C.2    Implementation and pretraining Details

We used the official source code provided by AttrMask, MoleBert, and SimSGT, retaining the exact same settings. Building upon this foundation, we introduced the GIBMS and SLG modules. The three additional hyperparameters for GIBMS were set to $t = 1$, $\beta = 0.01$, and $\tau = 0.1$, respectively. The four additional hyperparameters for SLG were set to $\tau_y = 0.1$, $\tau_p = 0.25$, $\alpha = 1$, and $n = 128$.

### C.3    Baselines

We now describe the details of our reported baseline methods:

- **InfoGraph** [32] conducts graph representation learning by maximizing the mutual information between graph-level representations and local substructures of various scales.

- **GPT-GNN** introduces a self-supervised attributed graph generation task to pre-train a GNN so that it can capture the structural and semantic properties of the graph. They factorize the likelihood of the graph generation into two components: 1) Attribute Generation and 2) Edge Generation.

- **ContextPred** [17] uses the embeddings of subgraphs to predict their context graph structures.

- **GraphLOG** [33] leverages clustering to construct hierarchical prototypes of graph samples. They further contrast each local instance with its corresponding higher prototype for contrastive learning.

- **Infomax** [38] learns node representations by maximizing the mutual information between the local summaries of node patches and the patches' graph-level global summaries.

- **G-Contextual** [29] views the prediction problem as a multi-class prediction task, where each class corresponds to one contextual property.

- **G-Motif** [29] formulates the prediction task as a multi-label classification problem, where each motif corresponds to one label.

- **AD-GCL** [34] applies adversarial learning for adaptive graph augmentation to remove the redundant information in graph samples.

- **JOAO** [48] proposes a framework to automatically search proper data augmentations for GCL.

- **SimGRACE** [40] take original graph as input and GNN model with its perturbed version as two encoders to obtain two correlated views for contrast. SimGRACE is inspired by the observation that graph data can preserve their semantics well during encoder perturbations while not requiring manual trial-and-errors, cumbersome search or expensive domain knowledge for augmentations selection.

- **GraphCL** [49] performs graph-level contrastive learning with combinations of four graph augmentations, namely node dropping, edge perturbation, subgraph cropping, and feature masking.

- **GraphMAE** [16] shows that a linear classifier is insufficient for decoding node types. It applies a GNN for decoding and proposes remask to decouple the functions of the encoder and decoder in the autoencoder.

Table 5: Transfer learning performance for molecular property prediction (regression) and drug target affinity (regression). **Bold** indicates the best performance.

| | Molecular Property Prediction (RMSE ↓) | | | | | Drug-Target Affinity (MSE ↓) | | |
| --- | --- | --- | --- | --- | --- | --- | --- | --- |
| | ESOL | Lipo | Malaria | CEP | Avg. | Davis | KIBA | Avg. |
| No Pre-train | $1.178_{\pm0.044}$ | $0.744_{\pm0.007}$ | $1.127_{\pm0.003}$ | $1.254_{\pm0.030}$ | 1.076 | $0.286_{\pm0.006}$ | $0.206_{\pm0.004}$ | 0.246 |
| ContextPred | $1.196_{\pm0.037}$ | $0.702_{\pm0.020}$ | $1.101_{\pm0.015}$ | $1.243_{\pm0.025}$ | 1.061 | $0.279_{\pm0.002}$ | $0.198_{\pm0.004}$ | 0.238 |
| AttrMask | $1.112_{\pm0.048}$ | $0.730_{\pm0.004}$ | $1.119_{\pm0.014}$ | $1.256_{\pm0.000}$ | 1.054 | $0.291_{\pm0.007}$ | $0.203_{\pm0.003}$ | 0.248 |
| JOAO | $1.120_{\pm0.019}$ | $0.708_{\pm0.007}$ | $1.145_{\pm0.010}$ | $1.293_{\pm0.003}$ | 1.066 | $0.281_{\pm0.004}$ | $0.196_{\pm0.005}$ | 0.239 |
| GraphMVP | $1.064_{\pm0.045}$ | $0.691_{\pm0.013}$ | $1.106_{\pm0.013}$ | $1.228_{\pm0.001}$ | 1.022 | $0.274_{\pm0.002}$ | $0.175_{\pm0.001}$ | 0.225 |
| Mole-BERT | $1.192_{\pm0.028}$ | $0.706_{\pm0.008}$ | $1.117_{\pm0.008}$ | $1.078_{\pm0.002}$ | 1.024 | $0.277_{\pm0.004}$ | $0.210_{\pm0.003}$ | 0.243 |
| SimSGT-G | $1.039_{\pm0.012}$ | $\mathbf{0.670_{\pm0.015}}$ | $1.090_{\pm0.013}$ | $1.060_{\pm0.011}$ | 0.965 | $0.263_{\pm0.006}$ | $0.144_{\pm0.001}$ | 0.204 |
| SimSGT-G-DyCC | $\mathbf{0.988_{\pm0.023}}$ | $0.672_{\pm0.016}$ | $\mathbf{1.082_{\pm0.012}}$ | $\mathbf{1.035_{\pm0.012}}$ | **0.944** | $\mathbf{0.256_{\pm0.003}}$ | $\mathbf{0.140_{\pm0.001}}$ | **0.198** |

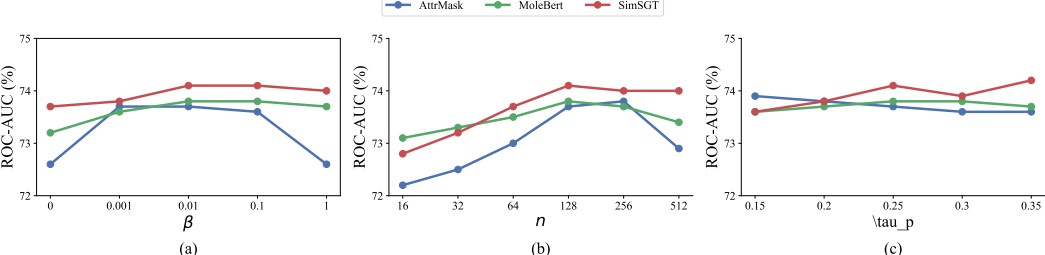

Figure 5: The impact of three hyperparameters $\beta$, $n$, and $\tau_p$.

- **GraphMVP** [24] uses a contrastive loss and a generative loss to connect the 2-dimensional view and 3-dimensional view of the same molecule, in order to inject the 3-dimensional knowledge into the 2-dimensional graph encoder.

- **MGSSL** [54] introdue a novel self-supervised motif generation framework for GNNs. First, for motif extraction from molecular graphs, they design a molecule fragmentation method that leverages a retrosynthesis-based algorithm BRICS and additional rules for controlling the size of motif vocabulary. Second, they design a general motif-based generative pretraining framework in which GNNs are asked to make topological and label predictions.

- **RGCL** [22] trains a rationale generator to identify the causal subgraph in graph augmentation. Each graph's causal subgraph and its complement are leveraged in contrastive learning.

- **Mole-BERT** [41] combines a contrastive learning objective and a masked atom modeling objective for MRL. Specifically, they observe that mask atom prediction is an overly easy pretraining task. Therefore, they employ a GNN tokenizer pretrained by VQ-VAEto generate more complex reconstruction targets for masked atom modeling.

## C.4 Broader Range of Downstream Tasks

We verified the efficacy of DyCC across a broader spectrum of downstream tasks and datasets, including four molecular property prediction regression tasks and two Drug-Target Affinity (DTA) regression tasks [27, 28]. DTA aims to predict the affinity scores between molecular drugs and target proteins. Following prior work [23], we pretrain SimSGT-DyCC on 50 thousand molecule samples from the GEOM dataset [6] and report the mean performances and standard deviations across three random seeds. We report the RMSE for the molecular property prediction datasets with scaffold splitting and report the MSE for the DTA datasets with random splitting. The results are summarized in Table 5. It is evident that SimSGT-DyCC surpasses the original version of SimSGT, achieving significant improvement over other baseline models. This suggests that DyCC can effectively enhance performance across a wider spectrum of downstream tasks.

## C.5 Additional experimental results

**Regression tasks** We verified the efficacy of DyCC across a broader spectrum of downstream tasks and datasets, including four molecular property prediction regression tasks and two Drug-Target

Table 6: Transfer learning ROC-AUC (%) scores on eight MoleculeNet datasets.The suffix "DyCC" implies the introduction of both the GIBMS and SLG modules.

| Dataset | Tox21 | ToxCast | Sider | ClinTox | MUV | HIV | BBBP | Bace | Avg. |
|---|---|---|---|---|---|---|---|---|---|
| No Pretrain | $74.6_{\pm0.4}$ | $61.7_{\pm0.5}$ | $58.2_{\pm1.7}$ | $58.4_{\pm6.4}$ | $70.7_{\pm1.8}$ | $75.5_{\pm0.8}$ | $65.7_{\pm3.3}$ | $72.4_{\pm3.8}$ | 67.0 |
| Mole-BERT | $76.2_{\pm0.5}$ | $63.9_{\pm0.3}$ | $61.4_{\pm1.9}$ | $75.1_{\pm3.0}$ | $77.4_{\pm2.1}$ | $77.5_{\pm1.0}$ | $66.8_{\pm1.5}$ | $78.9_{\pm0.9}$ | 72.2 |
| Mole-BERT DyCC | $76.3_{\pm0.5}$ | $64.4_{\pm0.5}$ | $61.4_{\pm0.9}$ | $78.9_{\pm2.4}$ | $78.6_{\pm1.9}$ | $77.7_{\pm0.9}$ | $70.8_{\pm0.6}$ | $82.2_{\pm0.9}$ | 73.8 |
| Mole-BERT DyCC + TMCL | $76.6_{\pm0.4}$ | $64.8_{\pm0.5}$ | $61.8_{\pm0.8}$ | $78.6_{\pm2.2}$ | $78.8_{\pm1.8}$ | $77.9_{\pm0.8}$ | $71.3_{\pm0.7}$ | $82.8_{\pm0.9}$ | 74.1 |
| SimSGT | $75.1_{\pm0.5}$ | $63.5_{\pm0.4}$ | $61.0_{\pm0.4}$ | $79.1_{\pm2.6}$ | $76.0_{\pm0.5}$ | $76.3_{\pm0.5}$ | $82.5_{\pm0.9}$ | $70.9_{\pm0.6}$ | 73.0 |
| SimSGT DyCC | $76.0_{\pm0.5}$ | $64.6_{\pm0.4}$ | $61.6_{\pm0.6}$ | $80.5_{\pm2.2}$ | $77.7_{\pm0.9}$ | $77.3_{\pm0.8}$ | $71.4_{\pm0.7}$ | $83.4_{\pm1.0}$ | 74.1 |
| SimSGT DyCC GraphTrans | $76.6_{\pm0.6}$ | $66.3_{\pm0.8}$ | $62.0_{\pm1.2}$ | $83.6_{\pm2.1}$ | $80.3_{\pm2.2}$ | $77.8_{\pm1.7}$ | $84.9_{\pm1.0}$ | $72.2_{\pm0.8}$ | 75.5 |

Table 7: Mean Average Error (MAE) performanceon the QM datasets.

| | QM7 | QM8 | QM9 |
|---|---|---|---|
| #Tasks | 1 | 12 | 12 |
| GraphCL | $80.4_{\pm3.3}$ | $0.0200_{\pm0.0004}$ | $5.76_{\pm0.37}$ |
| GraphMAE | $78.4_{\pm2.3}$ | $0.0190_{\pm0.0003}$ | $5.84_{\pm0.16}$ |
| Mole-BERT | $79.8_{\pm2.6}$ | $0.0190_{\pm0.0003}$ | $5.75_{\pm0.16}$ |
| Mole-BERT DyCC | $78.7_{\pm2.2}$ | $0.0188_{\pm0.0003}$ | $5.70_{\pm0.16}$ |
| SimSGT | $78.8_{\pm2.2}$ | $0.0189_{\pm0.0004}$ | $5.73_{\pm0.18}$ |
| SimSGT DyCC | $\mathbf{77.6_{\pm1.8}}$ | $\mathbf{0.0180_{\pm0.0003}}$ | $\mathbf{5.60_{\pm0.21}}$ |

Affinity (DTA) regression tasks [27, 28]. DTA aims to predict the affinity scores between molecular drugs and target proteins. Following prior work [23], we pretrain SimSGT-DyCC on 50 thousand molecule samples from the GEOM dataset [6] and report the mean performances and standard deviations across three random seeds. We report the RMSE for the molecular property prediction datasets with scaffold splitting and report the MSE for the DTA datasets with random splitting. The results are summarized in Table 5. It is evident that SimSGT-DyCC surpasses the original version of SimSGT, achieving significant improvement over other baseline models. This suggests that DyCC can effectively enhance performance across a wider spectrum of downstream tasks.

**Additional Modules of SimSGT and MoleBert**  In the main paper, for fair comparison, we excluded the GraphTrans variant of SimSGT and the TMCL proxy task of MoleBert. We provide the complete version in Table 6, and the results indicate that restoring these strategies further improves the model's performance. This suggests that DyCC can work in conjunction with other enhancements to MGAEs.

**Quantum chemistry property prediction.**  We report performances of predicting the quantum chemistry properties of molecules. We divide the downstream datasets by scaffold split. Specifically, we attach a two-layer MLP after the pretrained molecule encoders and fine-tune the models for property prediction. We report average performances and standard deviations across 10 random seeds. The performances are reported in Table 7. We observe a consistent enhancement of pre-trained model performance by DyCC.

## C.6  Hyperparameter experiments

Here, we explore several crucial hyperparameters of DyCC. The first one is $\beta$, which controls the trade-off between prediction and compression in our final objectives. As shown in Fig 5(a), there exists an optimal point at $\beta = 0.01$ in terms of model performance, indicating the trade-off between prediction and information compression. Setting a larger $\beta = 1$ encourages aggressive information compression, leading to difficulties in capturing the core subgraph related to the target task. Conversely, decreasing $\beta$ encourages the model to retain the original information of the given graph structure. In an extreme case, i.e., when $\beta = 0$, the model focuses solely on the prediction term, potentially leading to a lack of generalization ability. The second parameter is the number of prototypes $n$. We find that as $n$ increases from 16 to 128, the pretraining performance gradually improves, while using more prototypes has little effect. Therefore, $n = 128$ is a suitable choice. Lastly, $\tau_y$ and $\tau_p$ control the soft label sharpening level. We always choose $\tau_y < \tau_p$ to encourage

sharper target predictions, implicitly guiding the model to produce confident low-entropy predictions. We fix $\tau_y = 0.1$ and set $\tau_p$ to $\{0.15, 0.2, 0.25, 0.3\}$. Experimental results show that different models have different optimal values of $\tau_p$, which may be due to the different tokenizer types.

# D  Algorithm for GIBMS and SLT

Algorithm 1 and Algorithm 2 provide a comprehensive description of the GIBMS algorithm and the SLG process, respectively.

---

**Algorithm 1** The training process of GIBMS

---

1: **Input:** Unlabeled molecular pre-trained graph dataset $\mathcal{D}_1 = \{\mathcal{G}_1, \mathcal{G}_2, \cdots\}$, GNN encoder $\Phi$, and node importance evaluation MLP $\mathcal{M}$.
2: Initialize parameters of $\Phi$ and $\mathcal{M}$.
3: **for** each graph $\mathcal{G}$ in $\mathcal{D}_1$ **do**
4:     Encode $\mathcal{G}$ into node representations: $\mathbf{H} = \Phi(\mathcal{G})$.
5:     Generate a sampling probability for each node: $p = \mathcal{M}(\mathbf{H})$.
6:     Apply the Gumbel-Sigmoid function to sample $\lambda$ from $p$ based on Eq. (7).
7:     Inject noise into $\mathbf{H}$ to obtain $\hat{\mathbf{H}}$ based on Eq. (6),
8:     Compute the unsupervised prediction loss based on Eq. (11).
9:     Compute the loss for the compression term based on Eq. (12).
10:     Calculate the total loss for the first stage based on Eq. (13).
11:     Perform backpropagation to optimize the training objective.
12: **end for**
13: **Return:** The well trained $\Phi$ and $\mathcal{M}$ jointly constitute the GIBMS module $\mathcal{M}(\Phi(\mathcal{G}))$.

---

---

**Algorithm 2** Soft Label Generator

---

1: **Input:** Unlabeled molecular pre-trained graph dataset $\mathcal{D}_1 = \{\mathcal{G}_1, \mathcal{G}_2, \cdots\}$, the pre-trained GIBMS model $P(\Phi(\mathcal{G}))$, GNN encoder $\Phi'$ for MGAEs, and learnable prototypes matrix $\mathbf{Q}$.
2: Initialize the parameters of $\Phi'$ and $\mathbf{Q}$.
3: **for** each graph $\mathcal{G}$ in $\mathcal{D}_1$ **do**
4:     Compute the sampling probability for each node as $p = \text{Sigmoid}(\mathcal{M}(\mathbf{H}))$.
5:     Sample a set of important nodes $V_{\text{mask}} = \{V_i \mid V_i \sim \text{Bernoulli}(1 - p_i), i = 1, 2, \ldots, N\}$.
6:     Replace the nodes in $V_{\text{mask}}$ within graph $\mathcal{G}$ with a MASK token to obtain $\mathcal{G}_{\text{mask}}$.
7:     Obtain the node representations $\mathbf{H} = \Phi_2(\mathcal{G}_{\text{mask}})$.
8:     Compute the soft label assignments $s^p$ for all nodes by applying Eq. (15) to $\mathbf{H}$ and $\mathbf{Q}$.
9:     Compute the soft label assignments $s_y$ for all nodes by applying Eq. (14) to the node labels $y$ and $\mathbf{Q}$.
10:     Minimize the distance between $s_y$ and $s^p$ according to Eq. (17).
11:     Perform backpropagation to optimize the training objective.
12: **end for**
13: **Return:** The pre-trained GNN encoder $\Phi_2$ for various downstream tasks.

---

# E  Proof

## E.1  Proof of Eq. (10)

By adopting the common multi-view assumption [30, 42], we have:

$$
\begin{aligned}
I(\mathcal{G}_{\text{IB}}; \mathbf{Y}^{\text{sup}}) &= I\left(\mathbf{Z}_{\mathcal{G}_{\text{IB}}}^{\text{sup}}; \mathbf{Y}^{\text{sup}}\right) \\
&\geq I\left(\mathbf{Z}_{\mathcal{G}_{\text{IB}}}^{\text{ssl}}; \mathbf{Y}^{\text{sup}}\right) \\
&\geq I(\mathcal{G}_{\text{IB}}; \mathbf{Y}^{\text{sup}}) - \epsilon_{\text{info}}; \quad \epsilon_{\text{info}} > 0
\end{aligned}
$$

The proofs contain two parts [36]. The first one is showing the results for the supervised learned representations and the second one is for the self-supervised learned representations.

**Lemma 1 (Determinism)**  If $P\left(\mathbf{Z}_{\mathcal{G}_{\text{IB}}} \mid \mathcal{G}_{\text{IB}}\right)$ is Dirac, then the following conditional independence holds: $\mathbf{Y}^{\text{sup}} \perp \mathbf{Z}_{\mathcal{G}_{\text{IB}}} \mid \mathcal{G}_{\text{IB}}$ and $S \perp \mathbf{Z}_{\mathcal{G}_{\text{IB}}} \mid \mathcal{G}_{\text{IB}}$, inducing a Markov chain $\mathbf{S} \leftrightarrow \mathbf{Y}^{\text{sup}} \leftrightarrow \mathcal{G}_{\text{IB}} \rightarrow$

$\mathbf{Z}_{\mathcal{G}_{\mathrm{IB}}}$. When $\mathbf{Z}_{\mathcal{G}_{\mathrm{IB}}}$ is a deterministic function of $\mathcal{G}_{\mathrm{IB}}$, for any $A$ in the sigma-algebra induced by $\mathbf{Z}_{\mathcal{G}_{\mathrm{IB}}}$ we have $\mathbb{E}\left[\mathbf{1}_{\left[\mathbf{Z}_{\mathcal{G}_{\mathrm{IB}}}\in A\right]}\mid \mathcal{G}_{\mathrm{IB}},\{\mathbf{Y}^{\mathrm{sup}},\mathbf{S}\}\right]=\mathbb{E}\left[\mathbf{1}_{\left[\mathbf{Z}_{\mathcal{G}_{\mathrm{IB}}}\in A\right]}\mid \mathcal{G}_{\mathrm{IB}},\mathbf{S}\right]=\mathbb{E}\left[\mathbf{1}_{\left[\mathbf{Z}_{\mathcal{G}_{\mathrm{IB}}}\in A\right]}\mid \mathcal{G}_{\mathrm{IB}}\right]$, which implies $\mathbf{Y}^{\mathrm{sup}}\perp \mathbf{Z}_{\mathcal{G}_{\mathrm{IB}}}\mid \mathcal{G}_{\mathrm{IB}}$ and $\mathbf{S}\perp \mathbf{Z}_{\mathcal{G}_{\mathrm{IB}}}\mid G_{\mathrm{IB}}$.

**Supervised Learned Representations** Adopting Data Processing Inequality [9] in the Markov chain $\mathbf{S}\leftrightarrow\mathbf{Y}^{\mathrm{sup}}\leftrightarrow\mathcal{G}_{\mathrm{IB}}\to\mathbf{Z}_{\mathcal{G}_{\mathrm{IB}}}$ , $I\left(\mathbf{Z}_{\mathcal{G}_{\mathrm{IB}}};\mathbf{Y}^{\mathrm{sup}}\right)$ is maximized at $I(\mathcal{G}_{\mathrm{IB}};\mathbf{Y}^{\mathrm{sup}})$. Since the supervised learned representations $\mathbf{Z}_{\mathcal{G}_{\mathrm{IB}}}^{\mathrm{sup}}$ maximize $I\left(\mathbf{Z}_{\mathcal{G}_{\mathrm{IB}}};\mathbf{Y}^{\mathrm{sup}}\right)$ , we conclude $I\left(\mathbf{Z}_{\mathcal{G}_{\mathrm{IB}}}^{\mathrm{sup}};\mathbf{Y}^{\mathrm{sup}}\right)=I(\mathcal{G}_{\mathrm{IB}};\mathbf{Y}^{\mathrm{sup}})$.

**Self-supervised Learned Representations** First, we have

$$
\begin{aligned}
I\left(\mathbf{Z}_{\mathcal{G}_{\mathrm{IB}}};\mathbf{S}\right)&=I\left(\mathbf{Z}_{\mathcal{G}_{\mathrm{IB}}};\mathbf{Y}^{\mathrm{sup}}\right)\\
&\quad -I\left(\mathbf{Z}_{\mathcal{G}_{\mathrm{IB}}};\mathbf{Y}^{\mathrm{sup}}\mid\mathbf{S}\right)+I\left(\mathbf{Z}_{\mathcal{G}_{\mathrm{IB}}};\mathbf{S}\mid T\right)\\
&=I\left(\mathbf{Z}_{\mathcal{G}_{\mathrm{IB}}};\mathbf{Y}^{\mathrm{sup}};\mathbf{S}\right)+I\left(\mathbf{Z}_{\mathcal{G}_{\mathrm{IB}}};\mathbf{S}\mid\mathbf{Y}^{\mathrm{sup}}\right)
\end{aligned}
$$

and

$$
\begin{aligned}
I(\mathcal{G}_{\mathrm{IB}};\mathbf{S})&=I(\mathcal{G}_{\mathrm{IB}};\mathbf{Y}^{\mathrm{sup}})\\
&\quad -I(\mathcal{G}_{\mathrm{IB}};\mathbf{Y}^{\mathrm{sup}}\mid\mathbf{S})+I(\mathcal{G}_{\mathrm{IB}};\mathbf{S}\mid\mathbf{Y}^{\mathrm{sup}})\\
&=I(\mathcal{G}_{\mathrm{IB}};\mathbf{Y}^{\mathrm{sup}};\mathbf{S})+I(\mathcal{G}_{\mathrm{IB}};\mathbf{S}\mid\mathbf{Y}^{\mathrm{sup}})
\end{aligned}
$$

By DPI in the Markov chain $\mathbf{S}\leftrightarrow\mathbf{Y}^{\mathrm{sup}}\leftrightarrow\mathcal{G}_{\mathrm{IB}}\to\mathcal{G}_{\mathcal{G}_{\mathrm{IB}}}$, we know

- $I\left(\mathbf{Z}_{\mathcal{G}_{\mathrm{IB}}};\mathbf{S}\right)$ is maximized at $I(\mathcal{G}_{\mathrm{IB}};\mathbf{S})$
- $I\left(\mathbf{Z}_{\mathcal{G}_{\mathrm{IB}}};\mathbf{S};\mathbf{Y}^{\mathrm{sup}}\right)$ is maximized at $I(\mathcal{G}_{\mathrm{IB}};\mathbf{S};\mathbf{Y}^{\mathrm{sup}})$
- $I\left(\mathbf{Z}_{\mathcal{G}_{\mathrm{IB}}};\mathbf{S}\mid\mathbf{Y}^{\mathrm{sup}}\right)$ is maximized at $I(\mathcal{G}_{\mathrm{IB}};\mathbf{S}\mid\mathbf{Y}^{\mathrm{sup}})$

Since the self-supervised learned representations $\mathbf{Z}_{\mathcal{G}_{\mathrm{IB}}}^{\mathrm{ssl}}$ maximize $I\left(\mathbf{Z}_{\mathcal{G}_{\mathrm{IB}}};\mathbf{S}\right)$, we have $I\left(\mathbf{Z}_{\mathcal{G}_{\mathrm{IB}}}^{\mathrm{ssl}};\mathbf{S}\right)=I(\mathcal{G}_{\mathrm{IB}};\mathbf{S})$. Hence $I\left(\mathbf{Z}_{\mathcal{G}_{\mathrm{IB}}}^{\mathrm{ssl}};\mathbf{S}\mid\mathbf{Y}^{\mathrm{sup}}\right)=I(\mathcal{G}_{\mathrm{IB}};\mathbf{S}\mid\mathbf{Y}^{\mathrm{sup}})$. Using the result $I\left(\mathbf{Z}_{\mathcal{G}_{\mathrm{IB}}}^{\mathrm{ssl}};\mathbf{S};\mathbf{Y}^{\mathrm{sup}}\right)=I(\mathcal{G}_{\mathrm{IB}};\mathbf{S};\mathbf{Y}^{\mathrm{sup}})$, we get

$$
\begin{aligned}
I\left(\mathbf{Z}_{\mathcal{G}_{\mathrm{IB}}}^{\mathrm{ssl}};\mathbf{Y}^{\mathrm{sup}}\right)&=I(\mathcal{G}_{\mathrm{IB}};\mathbf{Y}^{\mathrm{sup}})\\
&\quad -I(\mathcal{G}_{\mathrm{IB}};\mathbf{Y}^{\mathrm{sup}}\mid\mathbf{S})\\
&\quad +I\left(\mathbf{Z}_{\mathcal{G}_{\mathrm{IB}}}^{\mathrm{ssl}};\mathbf{Y}^{\mathrm{sup}}\mid\mathbf{S}\right)
\end{aligned}
$$

Now, we are ready to present the inequalities:

$I(\mathcal{G}_{\mathrm{IB}};\mathbf{Y}^{\mathrm{sup}})\geq I\left(\mathbf{Z}_{\mathcal{G}_{\mathrm{IB}}}^{\mathrm{ssl}};\mathbf{Y}^{\mathrm{sup}}\right)$ due to $I(\mathcal{G}_{\mathrm{IB}};\mathbf{Y}^{\mathrm{sup}}\mid\mathbf{S})\geq I\left(\mathbf{Z}_{\mathcal{G}_{\mathrm{IB}}}^{\mathrm{ssl}};\mathbf{Y}^{\mathrm{sup}}\mid\mathbf{S}\right)$ by DPI and $I\left(\mathbf{Z}_{\mathcal{G}_{\mathrm{IB}}}^{\mathrm{ssl}};\mathbf{Y}^{\mathrm{sup}}\right)\geq I(\mathcal{G}_{\mathrm{IB}};\mathbf{Y}^{\mathrm{sup}})-\epsilon_{\mathrm{info}}$ due to

$$
\begin{aligned}
I(\mathcal{G}_{\mathrm{IB}};&\mathbf{Y}^{\mathrm{sup}})-I(\mathcal{G}_{\mathrm{IB}};\mathbf{Y}^{\mathrm{sup}}\mid\mathbf{S})+I\left(\mathbf{Z}_{\mathcal{G}_{\mathrm{IB}}}^{\mathrm{ssl}};\mathbf{Y}^{\mathrm{sup}}\mid\mathbf{S}\right)\\
&\geq I(\mathcal{G}_{\mathrm{IB}};\mathbf{Y}^{\mathrm{sup}})\\
&\geq I(\mathcal{G}_{\mathrm{IB}};\mathbf{Y}^{\mathrm{sup}})-\epsilon_{\mathrm{info}}
\end{aligned}\tag{18}
$$

where $I(\mathcal{G}_{\mathrm{IB}};\mathbf{Y}^{\mathrm{sup}}\mid\mathbf{S})\leq\epsilon_{\mathrm{info}}$ by the redundancy assumption.

### E.2 Proof of Eq. (12)

We derive the upper bound of $I\left(\mathcal{G};\mathcal{G}_{\mathrm{IB}}\right)$ by introducing the variation approximation $q\left(\mathcal{G}_{\mathrm{IB}}\right)$ of distribution $p\left(\mathcal{G}_{\mathrm{IB}}\right)$ :

$$
\begin{aligned}
I\left(\mathcal{G};\mathcal{G}_{\mathrm{IB}}\right)&=\mathbb{E}_{\mathcal{G},\mathcal{G}_{\mathrm{IB}}}\left[\log\frac{p_{\phi}\left(\mathcal{G}_{\mathrm{IB}}\mid\mathcal{G}\right)}{p(\mathcal{G})}\right]\\
&=\mathbb{E}_{\mathcal{G},\mathcal{G}_{\mathrm{IB}}}\left[\log\frac{p_{\phi}\left(\mathcal{G}_{\mathrm{IB}}\mid\mathcal{G}\right)}{q(\mathcal{G}_{\mathrm{IB}})}\right]\\
&\quad -\mathbb{E}_{\mathcal{G}_{\mathrm{IB}},\mathcal{G}}\left[KL\left(p\left(\mathcal{G}\right)\|q\left(\mathcal{G}_{\mathrm{IB}}\right)\right)\right]
\end{aligned}
$$

According to the non-negativity of KL divergence, we have:

$$
I\left(\mathcal{G}_{\mathrm{IB}},\mathcal{G}\right)\leq\mathbb{E}_{\mathcal{G}}\left[KL\left(p_{\phi}\left(\left(\mathcal{G}_{\mathrm{IB}}\mid\mathcal{G}\right)\|q\left(\mathcal{G}_{\mathrm{IB}}\right)\right)\right]\right.
$$

We assume that $q\left(\mathcal{G}_{\mathrm{IB}}\right)$ is obtained by aggregating the node representations in a fully perturbed graph. The noise $\epsilon \sim \mathcal{N}\left(\mu_{\mathbf{H}}, \sigma_{\mathbf{H}}^2\right)$ is sampled from a Gaussian distribution where $\mu_{\mathbf{H}}$ and $\sigma_{\mathbf{H}}^2$ are mean and variance of $\mathbf{H}$. Choosing sum pooling as the aggregatiion function, since the summation of Gaussian distributions is a Gaussian, we have the following form:

$$q\left(\mathcal{G}_{\mathrm{IB}}\right) = \mathcal{N}\left(N\mu_{\mathbf{H}}, N\sigma_{\mathbf{H}}^2\right)$$

Then for $p_\phi\left(\mathcal{G}_{\mathrm{IB}} \mid \mathcal{G}\right)$, we have the following equation:

$$\mathcal{N}\left(N\mu_{\mathbf{H}} + \sum_{j=1}^{N} \lambda_j \mathbf{H}_j - \sum_{j=1}^{N} \lambda_j \mu_{\mathbf{H}}, \sum_{j=1}^{N} (1 - \lambda_j)^2 \sigma_{\mathbf{H}}^2\right)$$

Finally, we have following inequality:

$$I\left(\mathcal{G}_{\mathrm{IB}}, \mathcal{G}\right) \leq \mathbb{E}_{\mathcal{G}}\left[-\frac{1}{2}\log A + \frac{1}{2N}A + \frac{1}{2N}B^2\right] + C$$

where $A = \sum_{j=1}^{N} (1 - \lambda_j)^2$ and $B = \frac{\sum_{j=1}^{N} \lambda_j (\mathbf{H}_j - \mu_{\mathbf{H}})}{\sigma_{\mathbf{H}}}$.

