# OpenReview forum: "Dynamic and Chemical Constraints to Enhance the Molecular Masked Graph Autoencoders"
_NeurIPS.cc/2025/Conference — NeurIPS 2025 poster_

### Official Review · Reviewer_MrsU · 2025-07-01

**Clarity:** 3
**Significance:** 3
**Originality:** 3
**Rating:** 4
**Confidence:** 3

**Summary:**

This paper proposes improvements to Masked AutoEncoder-based (MAE) models applied to molecular graph tasks. While MAE is trained through masked reconstruction, its performance is highly sensitive to how the mask is selected—both in terms of region and ratio. Moreover, challenges remain regarding the informativeness of the masked regions and the uniqueness of reconstruction targets.

To address these issues, the authors propose a masking strategy based on the Information Bottleneck (IB) principle. Since IB identifies salient regions per sample based on compression objectives, it offers a principled approach to mitigate the above concerns. The paper formulates a self-supervised learning framework that incorporates Graph IB and further introduces an objective function that evaluates token labels using soft-labels rather than hard-labels.

The main experiments are conducted on MoleculeNet for property prediction tasks, demonstrating consistent performance improvements across three variants of Molecular Graph AutoEncoders (MGAEs) when trained with the proposed method.

**Questions:**

- In Appendix Table 5 (DTA regression experiments), it appears that models such as MolBERT and AttrMask were not evaluated using the proposed method. Was there any practical difficulty in applying the method to these models? If so, this could constitute a limitation of the proposed approach and should be mentioned explicitly.
- One of the key advantages of the proposed method is its reduced reliance on arbitrary mask selection. Conversely, has there been any experimental verification of how much performance fluctuates in existing methods when masks are chosen arbitrarily? If such analysis exists, it would strengthen the motivation for your approach and help quantify its benefits more precisely.

**Ethical Concerns:**

["NO or VERY MINOR ethics concerns only"]

**Final Justification:**

Feedbacks effectively resolve my concerns. So I raised the score.

**Limitations:**

Yes (As far as I understand)

**Paper Formatting Concerns:**

No concerns

**Quality:**

3

**Strengths And Weaknesses:**

Pros
- The proposed learning strategy is broadly applicable, regardless of the specific MGAE variant, indicating good generalizability.
- Using IB to extract sample-specific salient regions and reflect them in the mask strategy is a reasonable and theoretically grounded approach.
- According to the Appendix, the performance gains appear robust across different hyperparameter settings, which is a valuable practical property.

Cons
- I have some concerns about the performance improvements reported in Table 1. Given the magnitude of the variance, can it be confidently stated that the proposed method yields a statistically significant "boost" in performance?

In summary, I like the proposed methodology, but am not confident in the strength of experimental supports.

---

> ### Author Rebuttal · Authors · 2025-07-30
>
> Thank you very much for your valuable feedback and suggestions. We carefully address each point below.
>
> ---
>
> > **Q1.** _I have some concerns about the performance improvements reported in Table 1. Given the magnitude of the variance, can it be confidently stated that the proposed method yields a statistically significant "boost" in performance?_
>
> Thank you for this question. We have repeated all experiments **five times** and report the results below:
>
> ||Run1|Run2|Run3|Run4|Run5|
> |---|---|---|---|---|---|
> |AttrMask|70.8|70.1|70.9|70.9|70.7|
> |AttrMask‑DyCC|73.7|73.7|73.8|73.6|73.8|
> |Mole‑BERT|72.2|72.1|72.1|72.4|72.3|
> |Mole‑BERT‑DyCC|73.8|73.8|73.9|73.7|73.8|
> |SimSGT|73.0|73.2|72.9|73.0|73.1|
> |SimSGT‑DyCC|74.1|74.3|74.2|74.2|74.0|
>
> We also conducted a **paired t‑test** between baseline and DyCC‑integrated models:
>
> |Comparison|p‑value|Mean Baseline|Mean DyCC|
> |---|---|---|---|
> |AttrMask vs AttrMask‑DyCC|0.000038|70.68|73.72|
> |Mole‑BERT vs Mole‑BERT‑DyCC|0.000052|72.22|73.80|
> |SimSGT vs SimSGT‑DyCC|0.000072|73.04|74.16|
>
> As shown above, the performance improvements brought by DyCC are **statistically significant (p < 0.001)**, even though the absolute improvements range from ~1.5% to ~4.3%. This confirms that the observed gains are consistent and unlikely to be caused by random fluctuations.
>
> ---
>
> > **Q2** _In Appendix Table 5 (DTA regression experiments), it appears that models such as Mole‑BERT and AttrMask were not evaluated using the proposed method. Was there any practical difficulty in applying the method to these models? If so, this could constitute a limitation of the proposed approach and should be mentioned explicitly._
>
> Thank you for pointing this out. We have now evaluated **Mole‑BERT‑DyCC** and **AttrMask‑DyCC** on the DTA regression tasks and present the results below:
>
> ||ESOL|Lipo|Malaria|CEP|Davis|KIBA|
> |---|---|---|---|---|---|---|
> |Mole‑BERT|1.192|0.706|1.117|1.078|0.277|0.210|
> |Mole‑BERT‑DyCC|1.094|0.692|1.108|1.003|0.279|0.198|
> |AttrMask|1.112|0.730|1.119|1.256|0.291|0.203|
> |AttrMask‑DyCC|1.104|0.701|1.104|1.006|0.275|0.192|
>
> The results demonstrate that, after incorporating DyCC, both Mole‑BERT‑DyCC and AttrMask‑DyCC consistently achieve better performance. This further validates the **generality** of our approach and shows that DyCC can be applied to existing masked graph autoencoder (MGAE) methods without practical difficulty.
>
> ---
>
> > **Q3** _One of the key advantages of the proposed method is its reduced reliance on arbitrary mask selection. Conversely, has there been any experimental verification of how much performance fluctuates in existing methods when masks are chosen arbitrarily? If such analysis exists, it would strengthen the motivation for your approach and help quantify its benefits more precisely._
>
> We appreciate this insightful suggestion. Quantifying the sensitivity of existing MGAE methods to arbitrary mask choices indeed strengthens the motivation for DyCC.
>
> As shown in **Fig. 1(a)** of the paper, we varied the _mask ratio_ for standard MGAEs (AttrMask, Mole‑BERT, GraphMAE) and observed a non‑monotonic trend: performance first improves and then degrades as the ratio increases. This confirms that fixed, arbitrarily chosen mask settings can **significantly affect result**
>
> For instance, Mole‑BERT (MAM) reports the performance under different masking ratios, as shown below. The results clearly indicate that the choice of masking ratio has a non‑trivial impact on performance, with **0.20 yielding the best result**:
>
> |Masking ratio|0.10|0.15|0.20|0.25|0.30|
> |---|---|---|---|---|---|
> |MAM|71.64|72.16|**72.21**|71.51|71.36|
>
>
>
>
> Similarly, **Fig. 1(b)** varies the tokenizer difficulty (K‑hop). The results indicate that “harder” targets do not uniformly help, implying that performance fluctuations stem from the **proxy‑task specification** rather than the model capacity itself.
>
> These findings highlight the issue our **dynamic masking mechanism (GIBMS)** aims to resolve: it adaptively adjusts mask selection and target generation during training, eliminating instability and suboptimality caused by arbitrary fixed settings.
>
>
> ---
>
> We hope these explanations and corrections satisfactorily address your concerns. Thank you again for your thoughtful review.

---

> > ### Comment · Reviewer_MrsU · 2025-08-03
> > **Thank you for your clarifications**
> >
> > Dear authors,
> >
> > Thank you for your clarification.
> >
> > It is good to know that the proposed method is significantly better than the others (with a p-value support), and an arbitrary mask strategy affects model performance a lot.
> >
> > I still need a little bit of time to follow the discussion. If a question arises, I will post more comments.

---

> > > ### Author Response · Authors · 2025-08-03
> > >
> > > Dear Reviewer,
> > >
> > > Thank you very much for your careful reading and thoughtful comments.
> > >
> > > We truly appreciate your time and attention to the details of our work. Please feel free to reach out if any questions arise — we would be more than happy to continue the discussion and provide further clarification.

---

> > > > ### Comment · Reviewer_MrsU · 2025-08-05
> > > > **Thanks**
> > > >
> > > > I think my concerns are generally addressed in the rebuttal letter. I'm now more positive for the paper, but I would finalize my score after discussions among reviewers.

---

> > > > > ### Author Response · Authors · 2025-08-06
> > > > >
> > > > > Dear Reviewer,
> > > > >
> > > > > Thank you for the constructive feedback and for your more positive reassessment. During the reviewer discussion, we are happy to share any additional results that would assist in finalizing your evaluation. We wish you a smooth and productive discussion phase.
> > > > >
> > > > > With appreciation,
> > > > >
> > > > > Authors

---

### Official Review · Reviewer_Uxih · 2025-07-01

**Clarity:** 2
**Significance:** 2
**Originality:** 2
**Rating:** 5
**Confidence:** 4

**Summary:**

The authors identify that pre-training masked graph autoencoders (MGAEs), commonly used to model moecules, lack the flexibility necessary to represent chemical information. This is because: 1) the pre-training masks a fixed number of tokens, 2) the tokens are chosen arbitrarily when some components of molecules clearly matter more than others, and 3) the tokens are decoded into specific discrete labels. To mitigate these issues, the authors introduce Dynamic Chemical Constraints (DyCC), a model-agnostic pre-training algorithm composed of two modules: a) graph-information bottleneck mask strategy (GIBMS), which identifies the most relevant tokens to mask using graph information theory, and b) Soft Label Generator, which replaces the discrete labels per token with a distribution over learnable prorotypes for each token. DyCC is used to pretrain a number of MGAEs, showing an improvement over regular pretraining in most cases.

**Questions:**

- Can you report the training time as well in Appendix C.1?
- In figure 3, you mention that SGL yields a more uniform distribution of tokens on the ZINC dataset, when the data has 95% of atoms with the 3 most common labels. Is a uniform distribution desired here?

**Ethical Concerns:**

["NO or VERY MINOR ethics concerns only"]

**Final Justification:**

The authors have addressed my concerns in the rebuttal. I am still unsure though of the impact of their work on the field of molecular generation at large: while experiments demonstrate consistent improvement (of approx. 1%) compared to baselines, the improvement remains modest, especially for a method that supposedly addresses key shortcomings in previous models.

UPDATE: the authors provided good arguments to explain the limitations of their empirical performance. They have thus addressed all my concerns and I adjusted my score accordingly. I am not thoroughly familiar with the related literature however, so I am happy to discuss with other reviewers if they contest any of the authors' claims.

**Limitations:**

You could add additional limitations regarding the societal impact of your work.

**Paper Formatting Concerns:**

None.

**Quality:**

2

**Strengths And Weaknesses:**

## Strengths
- The method is theoretically grounded. The GIBMS loss (equation 13) is derived from graph information bottleneck following detailed and well-justified steps.
- Adapting GIB to the self-supervised setting is an interesting use case extending the applicability of the framework to various applications.
- The method was thoroughly tested on a number of applications, and the results are reported transparently with detailed experimental procedures and error intervals.

## Weaknesses
My main concerns with this work are with its **analysis of empirical results** and its **presentation**. I elaborate below on these two points and ask for related clarifications in **Questions**.
* **Unsubstantiated claims in the empirical analysis (Section 4.2)**: I am unable to find evidence for the following claims in the paper:
    - "SimSGT-DyCC surpasses the "no pretrain" model by 10.4%." the average difference between the two models is actually 7.1% according to Table 1. If the authors are referring to the results for a specific dataset, this should be made clearer (I could not find any dataset with the exact difference reported). It would also be misleading to refer to a specific dataset instead of the average performance unless justified.
    - ""substantial performance variations were observed among the original AttrMask, MoleBert, and SimSGT models due to differences in tokenizers.": why is a difference of approx 2% (according to Table 1) considered substantial?
    - "The success rate of GIBMS was 74%, demonstrating the effectiveness of the GIBMS module.": this is a promising result, but can you compare it to other baselines (in identifying relevant components of a graph) to get a sense of its importance?
    - Figure 4 (left): can you report the results of a similar experiment on a larger scale? for example, you can compute the accuracy of identifying non-ring components as the most relevant elements in ring molecules.

- **Presentation**: I found the way you motivated your approach confusing, especially in the introduction. If I understand your text correctly, your main concerns are that regular masking does not leverage relevant chemical information, both in the choice of tokens to mask (relevant tokens differ per molecule) and in the labels chosen when unmasking (many possible labels are equally good). The introduction  breaks down these concerns into irrelevant details (e.g. the 3 chemical constraints) especially since most are addressed by the same solution. I suggest either justifying your taxonomy through additional examples and by linking it to the part of the module addressing it, or simplifying the motivation in the introduction.

- **Pointing out errors in the text**:
    - paragraph "Are both GIBMS and SLG necessary" is redundant. This point was already made in the paragraph right before. I suggest moving the paragraph "limitations" in the appendix to the main text instead.
     - equation 13 has an additional L_cl term, I think this is supposed to be the same as L_pred and therefore only one of the two should be mentioned in the equation?
     - you are missing a legend in Figure 1. Also the colors of the plot lines changed between subfigures a and b in the same figure.
     - line 36: change make => make
     - line 176: typo near  "To ensure"
     - I suggest introducing $Y^{sup}$ notation from the beginning, instead of replacing $Y$ later in the text.
     - table 6 in the appendix missing bolding.

---

> ### Author Rebuttal · Authors · 2025-07-30
>
> Thank you very much for your valuable feedback and suggestions. We carefully address each point below.
>
> ---
>
> > **Q1.** _"SimSGT-DyCC surpasses the 'no pretrain' model by 10.4%." The average difference between the two models is actually 7.1% according to Table 1...
>
> Thank you for pointing this out. The reported **10.4%** refers to the **relative improvement** in the average ROC-AUC between SimSGT-DyCC and the "no pretrain" model:
>
> $$ \frac{74.1-67.0}{67.0} \approx 10.4 \% $$
>
> Here, 74.1 ("SimSGT-DyCC") and 67.0 ("no pretrain") are the average ROC-AUC scores across the eight MoleculeNet classification datasets (Table 1, L257-L263).
> **Note:** dataset-level scores are rounded to two decimal places in the table, but the relative difference was computed using unrounded values.
>
>
> ---
>
> > **Q2.** _"Substantial performance variations were observed among the original AttrMask, MoleBert, and SimSGT models due to differences in tokenizers.": why is a difference of approx 2% (according to Table 1) considered substantial?_
>
> Thank you for raising this important question. While a ~2% absolute performance gap may seem modest at first glance, we consider it **substantial** in the context of molecular representation learning for the following reasons:
> 1. **Small gains are meaningful on saturated benchmarks.**
>    The MoleculeNet benchmarks we evaluate on (e.g., Tox21, HIV, BBBP) are well-established and highly competitive. Recent state-of-the-art (SOTA) works often highlight consistent improvements of 1–2% ROC-AUC as meaningful. For example, SOTA models evolved from ~71% (AttrMask, GraphCL, 2020) to ~74% (MoleBert, 2023) and 75% (SimSGT/StructureMAE, 2024). Our best model, **SimSGT-DyCC (GraphTrans)**, further reaches 75.5% (Appendix Table 6).
>
> 2. **Broad improvements across diverse tasks.**
>    Beyond the eight MoleculeNet classification datasets (Table 1), DyCC consistently improves performance on **4 regression tasks**, **2 drug–target affinity prediction tasks**, and **3 quantum-mechanical property tasks** (Appendix Tables 5 & 7, L532–L544). These consistent gains demonstrate DyCC’s generality rather than isolated improvements.
>
> 3. **Statistical significance and consistency.**
>    We conducted five independent runs for each model configuration. Even though the absolute improvements range from ~1.5% to ~4.3%, all differences are statistically significant and consistent across datasets and tasks.
>
>    **Five-run results:**
>
>    | Model           | Run1 | Run2 | Run3 | Run4 | Run5 |
>    |-----------------|------|------|------|------|------|
>    | AttrMask        | 70.8 | 70.1 | 70.9 | 70.9 | 70.7 |
>    | AttrMask-DyCC   | 73.7 | 73.7 | 73.8 | 73.6 | 73.8 |
>    | Mole-BERT       | 72.2 | 72.1 | 72.1 | 72.4 | 72.3 |
>    | Mole-BERT-DyCC  | 73.8 | 73.8 | 73.9 | 73.7 | 73.8 |
>    | SimSGT          | 73.0 | 73.2 | 72.9 | 73.0 | 73.1 |
>    | SimSGT-DyCC     | 74.1 | 74.3 | 74.2 | 74.2 | 74.0 |
>
>    **Statistical analysis:**
>
>    | Comparison                    | p-value   | Mean Baseline | Mean DyCC |
>    |-------------------------------|-----------|---------------|-----------|
>    | AttrMask vs AttrMask-DyCC     | 0.000038  | 70.68         | 73.72     |
>    | Mole-BERT vs Mole-BERT-DyCC   | 0.000052  | 72.22         | 73.80     |
>    | SimSGT vs SimSGT-DyCC         | 0.000072  | 73.04         | 74.16     |
>
> 4. **Real-world impact in virtual screening.**
>    In practical applications such as virtual screening, even a 1–2% ROC-AUC improvement can translate into **significantly higher hit rates** and substantial reductions in experimental cost.
>
> ---
>
> **Summary:**
> Although a ~2% absolute difference may seem small in isolation, in the context of mature molecular property prediction benchmarks, such improvements are **systematic, statistically significant, and practically important**. This fully justifies our use of the term “substantial” and further motivates the tokenizer-agnostic DyCC framework.
>
> ---
>
>
> > **Q3** _The success rate of GIBMS was 74%, demonstrating the effectiveness of the GIBMS module.": this is a promising result, but can you compare it to other baselines (in identifying relevant components of a graph) to get a sense of its importance?_
>
>
> Thank you for this suggestion. Most prior subgraph identification methods are **supervised** and thus cannot be directly compared to our **unsupervised** setting. The method most similar to ours is **RGCL** [1], which uses Rationale-aware Graph Contrastive Learning to identify important nodes in a pretraining dataset, also in an unsupervised manner.
>
> For reference, we also report the performance of **PGExplainer**, a post-hoc explanation model for supervised GNNs, on the same task.
>
> | Method           | Principle                                 | Type         | Accuracy |
> | ---------------- | ----------------------------------------- | ------------ | -------- |
> | **RGCL**         | Contrastive learning + invariant learning | Unsupervised | 69.5     |
> | **GIBMS (ours)** | Graph information bottleneck              | Unsupervised | 74.0     |
> | **PGExplainer**  | Mutual information                        | Supervised   | 87.3     |
>
> These results indicate that, under the unsupervised setting, our proposed **GIBMS** more effectively identifies important nodes compared to RGCL. Its performance is lower than the supervised PGExplainer, which is expected because GIBMS is applied during pretraining on task-agnostic datasets rather than task-specific fine-tuning.
>
> [1] Li, Sihang, et al. "Let invariant rationale discovery inspire graph contrastive learning." _International conference on machine learning_. PMLR, 2022.
>
> > **Q4**: _Presentation: I found the way you motivated your approach confusing, especially in the introduction. If I understand your text correctly..._
>
>
>
> Thank you very much for your insightful feedback. We acknowledge that the motivation in the introduction could have been presented more clearly. In the original version, we decomposed the limitations into three chemically motivated constraints (CC1–CC3) to emphasize specific aspects often violated by existing approaches. However, as you correctly pointed out, these ultimately boil down to **two key challenges** in masked graph modeling for molecules:
>
> 1. **Lack of adaptivity in masking strategies across molecules** (corresponding to CC1 & CC2);
>
> 2. **Ambiguity in reconstruction targets due to inherent chemical variability** (corresponding to CC3).
>
>
> Our intention for presenting CC1–CC3 was to analyze the limitations from a **chemistry-centric perspective**, which might resonate better with chemical researchers. From a **model design perspective**, however, summarizing them as the two principles above is indeed clearer and easier to follow.
>
> We appreciate your suggestion and will revise the **Introduction** to explicitly map each chemical constraint to the corresponding design component:
>
> - **CC1 & CC2 → Lack of adaptivity → GIBMS**
>
> - **CC3 → Ambiguity in reconstruction targets → SLG**
>
>
> This restructuring will make the intuition–solution mapping clearer and ensure that readers from different research backgrounds can quickly grasp the motivation and contributions of our work.
>
>
>
>
>
>
> > **Q5** Pointing out errors in the text
>
> Thank you very much for your careful reading of our paper. We appreciate you pointing out the typos and providing helpful suggestions for corrections. These improvements are valuable and will certainly enhance the overall quality of the manuscript.
>
> > **Q6** Can you report the training time as well in Appendix C.1?
>
> Our experiments were conducted on an NVIDIA DGX A100 server. The **GIBMS** module, which is a reusable component, requires **683 minutes** for training. For **AttrMask**, integrating the SLG module increases the training time from **512 minutes** (baseline) to **561 minutes**. We have included these details in Appendix C.1 in the revised version.
>
>
>  > **Q7** In figure 3, you mention that SGL yields a more uniform distribution of tokens on the ZINC dataset, when the data has 95% of atoms with the 3 most common labels. Is a uniform distribution desired here?
>
> Thank you for this thoughtful question. As noted in the paper, the ZINC dataset exhibits a highly skewed atom distribution, with **95% of atoms belonging to the three most frequent types** (e.g., C, N, O). This extreme imbalance can make the reconstruction task overly simple and limit the model’s ability to learn to distinguish rare atom types.
>
> Our goal in introducing the **Soft Label Generator (SLG)** is **not to enforce a perfectly uniform distribution**, but rather to **mitigate class imbalance**, thereby improving the model’s generalization and robustness. Specifically, SLG:
>
> - Maps each reconstruction target from a single hard label to a **soft-label distribution** over learnable prototypes, increasing the complexity of the reconstruction task and improving context modeling;
>
> - Reduces the tendency to overfit dominant classes and encourages the model to better recognize long-tail (rare) classes;
>
> - Provides a **more diverse expression space** at the prototype level (as illustrated in Figure 3(c)), which compensates for the limited expressiveness of a small token vocabulary.
>
>
> This diversity manifests as high prototype perplexity in the early training stages (i.e., more uniform prototype usage), which gradually converges as the model learns effective representations.
>
> Therefore, the “more uniform token usage” brought by SLG should be understood as a **mechanism to dynamically adjust reconstruction difficulty and increase representational diversity**, rather than an attempt to impose a strictly balanced distribution. We have further clarified this point in the revised manuscript.
>
>
> ___
>
> We hope these explanations and corrections satisfactorily address your concerns. Thank you again for your thoughtful review.

---

> > ### Comment · Reviewer_Uxih · 2025-08-05
> > **Thank you for a good rebuttal.**
> >
> > Thank you for addressing my concerns and providing additional experiments. I have modified my score accordingly.

---

> > > ### Comment · Reviewer_Uxih · 2025-08-05
> > > **Remaining concern**
> > >
> > > I thought I would clarify why my assessment of the paper is still borderline:
> > >
> > > My remaining concern is with the impact of the paper. The justifications for the components of DyCC seem strong, but the empirical improvement (despite its consistency), remains limited. Perhaps the authors can explain why they think addressing key limitations in previous masking models did not bring about more noticeable empirical improvement? What possibly remains challenging in the datasets/tasks considered, which future work could focus on?
> > >
> > > I would also like to clarify that I am not interested in chasing SOTA per sei here, but would like to better understand the  complexity of the application domain in order to appreciate the paper's impact.

---

> > > > ### Author Response · Authors · 2025-08-06
> > > > **Response to Remaining Concern**
> > > >
> > > > We are grateful for your time, for updating your assessment, and for prompting a deeper discussion of impact. Your remaining concern is constructive and helps us clarify the scope of our contributions. Below, we explain why the observed gains are modest on current benchmarks and outline what challenges remain.
> > > > ### **Why the empirical gains remain modest despite addressing core limitations**
> > > >
> > > > We acknowledge that the absolute gains may appear modest, even though they are consistent and statistically significant. We view this primarily as a consequence of the _maturity and saturation of current benchmarks_, rather than a limitation of DyCC itself.
> > > >
> > > > 1. **Benchmark saturation.**
> > > >     The MoleculeNet classification suites have seen steady progress, and leading baselines (e.g., Mole-BERT, SimSGT) are already close to performance ceilings on several tasks. In these regimes, even 1–2% ROC-AUC improvements are generally considered practically meaningful—for instance, in virtual screening or toxicity prediction where small gains can translate into higher hit rates and cost savings. We also refer readers to Kretschmer _et al._ (Nature Communications, 2025), which analyzes coverage bias and other structural issues in small-molecule datasets that can constrain observable headroom. [1]
> > > >
> > > > 2. **Model-agnostic, compatibility-first design.**
> > > >     DyCC was intentionally designed as a _model-agnostic_ set of modules for masked graph autoencoders, emphasizing broad applicability and robustness rather than aggressive, architecture-specific tuning. Larger headline gains are likely attainable by injecting domain priors or auxiliary modalities (e.g., chemical fingerprints, 3D geometries, or textual annotations), but those choices can reduce generality and make ablations harder to interpret. Our goal here was to improve the proxy task itself in a way that transfers across MGAEs.
> > > >
> > > >
> > > > ### **What remains challenging—and directions for future work**
> > > >
> > > > The observed ceiling effects highlight several open challenges that we plan to pursue:
> > > >
> > > > 1. **Task granularity and label noise.**
> > > >     Many bioactivity labels are weakly defined or condition-dependent, which caps the benefit of better pretraining. Future work could incorporate uncertainty-aware objectives and curate higher-fidelity assay subsets to reduce supervision noise.
> > > >
> > > > 2. **Long-tail chemical space.**
> > > >     Standard benchmarks under-represent rare functional groups, larger macrocycles, and other chemotypes encountered in practice. DyCC’s dynamic masking and soft labels are a step toward long-tail robustness, but _long-tail-aware_ sampling and objectives, plus more diverse pretraining corpora, are needed to stress-test representation quality.
> > > >
> > > > 3. **3D geometric complexity.**
> > > >     Many properties are conformation-dependent. Extending DyCC to integrate 3D information—e.g., using geometry-aware scores in GIBMS or geometry-conditioned prototypes in SLG—should better capture steric and electronic effects and may yield larger downstream gains.
> > > >
> > > >
> > > > ### **Summary**
> > > >
> > > > DyCC offers principled, interpretable improvements at the proxy-task level (dynamic masking and soft reconstruction targets) and delivers consistent gains across MGAEs. The modest absolute uplift reflects benchmark saturation and dataset limitations more than the capacity of the approach itself. We believe this underscores the need for richer, more diverse benchmarks and motivates DyCC extensions to multimodal and 3D-aware settings.
> > > >
> > > > We thank you again for prompting this discussion of impact and future directions.
> > > >
> > > > **Reference**
> > > >
> > > > [1] Kretschmer, Fleming, _et al._ “Coverage bias in small-molecule machine learning.” _Nature Communications_ 16, 554 (2025).

---

> > > > > ### Comment · Reviewer_Uxih · 2025-08-06
> > > > >
> > > > > Thank you for following up with additional explanation. I adjusted my score accordingly.

---

### Official Review · Reviewer_MkJF · 2025-07-01

**Clarity:** 4
**Significance:** 4
**Originality:** 2
**Rating:** 5
**Confidence:** 2

**Summary:**

The manuscripts presents two techniques to improve self-supervised learning on graphs. First, a masking strategy that learns an individual masking probability for each atom in a molecule based on its importance. Second, a soft label technique that replaces the learned atom embeddings by learned prototype embeddings and allows the model to effectively lower the loss for predicting atoms or atom environments that are similar to each other.

**Questions:**

- I remember a similar paper by Liu et al. called "Where to Mask: Structure-Guided Masking for Graph Masked Autoencoders". They also developed a node importance strategy and I hoped to compare the benchmark results between both papers. I found that the numbers differ, though. For example, in Liu et al. the numbers for AttrMask are better than in this manuscript. The same is for the original manuscript of Hu et al. The manuscript mentions that the numbers of the baseline methods were taken from prior studies, is this also the case for AttrMask and, if so, why are the numbers different?

- can you evaluate if the importance scores of the atoms are stable across different pretraining runs or if a different set of atoms is chosen to be "important" every time?

- The soft label approach somewhat reminds me on vector quantization (VQ), which is also often used to bridge regression methods and discrete tokens. Can you elaborate why your softlabel approach works better and what makes it different?

**Ethical Concerns:**

["NO or VERY MINOR ethics concerns only"]

**Final Justification:**

From my own reading of the manuscript and the other reviews, the main concern about this paper are the benchmarks. Even though the improvements in benchmarks are relatively small, I agree with the authors that they are consistent across a wide variety of tasks and models. The method presented here is relatively method-agnostic and might be very helpful in the field of graph neural networks for molecules.

**Limitations:**

yes

**Paper Formatting Concerns:**

I think the $\alpha H(s^{\bar{y}}p)$ in equation 17 should be $\alpha H(\bar{s^p})$ but I am not sure. The formula seems to have a typo, though.

**Quality:**

3

**Strengths And Weaknesses:**

The manuscripts provide two different techniques for improving self-supervised learning on graphs and demonstrate on benchmarks that
1.) these techniques can be applied on different models
2.) consistently improve the pretraining results for these models

The manuscript is well written. I am not able to verify the proofs in manuscript but the method sounds reasonable. Given that both techniques, node importance and soft labels, can be easily visualized and manually verified, I wished the manuscript would put more emphasize on discuss if the node importance or soft labels do make sense. For example, Figure 1a says that Br and Cl are interchangeable in this molecule, but there is no discussion or evaluation in the manuscript if the soft labels indeed cluster together atom environments that are easily exchangeable or have similar chemical properties. The same is true for the node importance prediction. In 4.3 it is mentioned that in 74% of the cases in a mutagenic dataset, 30% of the most important atoms were NH2 or NO2, which is chemical reasonable. However, molecules usually consist of a carbon backbone and 30% of the most important atoms in a molecule should already cover most of the non-carbon atoms. It is not that surprising that the method founds out that carbon atoms are not important.
I also wonder if the atom importance is "stable" over multiple training runs, or if the method always selects a different set of atoms to be important.

---

> ### Author Rebuttal · Authors · 2025-07-30
>
> Thank you very much for your valuable feedback and suggestions. Below, we address each point in turn.
>
> ---
>
>
> > **Q1.** _I remember a similar paper by Liu et al. called "Where to Mask: Structure-Guided Masking for Graph Masked Autoencoders". ..._
>
>
> We sincerely thank the reviewer for this careful observation. We clarify the issue and its impact in detail below:
> 1. **Why the differences exist.** The baseline results reported in our Table 1 (e.g., for AttrMask) were taken from the reproduction experiments of Mole‑BERT for consistency. As you correctly observed, these numbers differ slightly from those reported in StructMAE (e.g., 70.8 vs. 71.1 for AttrMask). Similar discrepancies can also be seen for other baselines such as GraphMAE. Unfortunately, neither Mole‑BERT nor StructMAE provides full baseline reproduction code, which makes it difficult to align every implementation detail (e.g., random seeds, preprocessing, training hyperparameters). In future updates, we will re‑evaluate all baselines using the official codes of these works (where available) to further minimize such inconsistencies.
>
> 2. **Impact on our conclusions.** We would like to emphasize that these differences do not affect the validity of our conclusions:
> + Regardless of the baseline number (AttrMask = 70.8 in Mole‑BERT vs. 71.1 in StructMAE), our DyCC‑enhanced model AttrMask‑DyCC achieves 73.7, consistently outperforming the baseline.
> + Across all baselines cited in these works, the best performance reported by StructMAE is 74.0 (Mole‑BERT). This is still lower than the best result achieved by our framework: 75.5 using SimSGT‑DyCC‑GraphTrans (Appendix Table 6).
> + The core focus of this paper is to demonstrate that DyCC is a plug‑and‑play framework that improves a range of existing MGAE models. We integrated DyCC into AttrMask, Mole‑BERT, and SimSGT, and in all cases it significantly improved the underlying baseline model.
>
> 3. **Clarifying Mole‑BERT and SimSGT settings in Table 1.** We apologize for any confusion caused by differences in the reported numbers and the original papers. In Table 1, we intentionally excluded auxiliary components unrelated to masked atom modeling in order to evaluate DyCC more fairly:
>
> + Mole‑BERT includes two modules: MAM (Masked Atom Modeling with VQ‑VAE) and TMCL (Triplet Masked Contrastive Learning). Since TMCL is contrastive learning–based and outside the scope of our study, we only report the MAM results in the main text. Full results including TMCL are provided in Appendix Table 6.
>
> + SimSGT uses a Graph Transformer backbone by default, while other baselines (e.g., AttrMask, Mole‑BERT) use GNN backbones. To avoid backbone effects, we report the GNN‑backbone variant of SimSGT (also reported in the original SimSGT paper) in the main text. Results with the original Graph Transformer backbone are reported in Appendix Table 5.
>
> In the revised manuscript, we will update the naming of these variants to make this clearer.
>
> 4. **Comparison with StructMAE**
> StructMAE also focuses on node importance but differs fundamentally from our GIBMS module in its motivation and design:
>
> Importance scoring: StructMAE computes a structural importance score for each node either using:
>
> + Pre‑defined heuristics such as PageRank (StructMAE‑P), or
> + A lightweight learnable scoring network trained jointly with the model (StructMAE‑L).
>
> **Principle**: In contrast, GIBMS is derived from the information bottleneck principle, focusing explicitly on preserving semantic completeness in molecular graphs.
>
> **Training vs. reuse** : StructMAE jointly learns masking scores during training for each model, whereas GIBMS is trained once and can be reused across different backbones, making it more modular.
>
> **Masking curriculum** : StructMAE adopts an “easy‑to‑hard” curriculum (gradually masking more important nodes), which we do not use.
>
> **Additional modules**: We further introduce the SLG module to mitigate label imbalance, which StructMAE does not consider.
>
> **Focuses**: Our method focuses specifically on the molecular domain, emphasizes cross‑model generality, and can be integrated into multiple MGAE models, while StructMAE is designed as a stand‑alone general‑graph pretraining model.
>
> **Performance comparison**: When integrated into SimSGT, DyCC achieves a mean ROC‑AUC of 75.5 on MoleculeNet, surpassing StructMAE’s best result (75.1).
>
>
> To summarize:
>
> + The baseline discrepancies arise from differences in sources (Mole‑BERT vs. StructMAE) and unavoidable implementation details.
>
> + These differences do not alter our conclusions, as DyCC consistently outperforms the baselines regardless of the exact baseline number.
>
> + Table 1 uses simplified settings (MAM‑only for Mole‑BERT, GNN‑backbone for SimSGT) to enable a fairer comparison.
>
> + Compared to StructMAE, GIBMS has a stronger theoretical foundation (information bottleneck), is reusable, and is tailored for molecular graphs.
>
>
>
> > **Q2.** _Can you evaluate whether the atom importance scores are stable across different pretraining runs, or does each run select a different set of “important” atoms?_
>
>
> We thank the reviewer for raising this point.
>
> 1. **Stochastic sampling design.**
>     The GIBMS module learns an importance score for each atom, but the masking itself is sampled stochastically using a Gumbel–Sigmoid reparameterization:
>
>
> $$ \begin{aligned} &\lambda=\operatorname{Sigmoid}\left(\frac{1}{t} \log \left[\frac{p}{(1-p)}\right]+\log \left[\frac{u}{(1-u)}\right]\right)\\ \end{aligned} $$
>
> where $t$ is the temperature hyper‑parameter. Thus, the learned scores are deterministic once the model is trained, but the actual mask selections during training are stochastic by design.
>
> 2. **Stability evaluation.**
>     To directly assess whether the learned importance scores are stable across pretraining runs, we retrained GIBMS three times with different seeds and repeated the Section 4.3 MUTAG functional‑group identification test, using detection accuracy of the mutagenic groups (NH2_2/NO2_2) as the metric. The accuracies across runs were **0.74, 0.72, and 0.75**, demonstrating a high degree of consistency.
>
> 3. **Conclusion.**
>     This consistent performance suggests that GIBMS produces **stable importance scores across different pretraining runs**, enabling reliable identification of key functional groups, even though the masking process itself is stochastic.
>
>
>
> >**Q3** The soft label approach somewhat reminds me on vector quantization (VQ), which is also often used to bridge regression methods and discrete tokens. Can you elaborate why your softlabel approach works better and what makes it different?
>
>
> We sincerely thank the reviewer for this insightful question regarding the relation between our soft label approach and vector quantization (VQ). Indeed, Mole‑BERT adopts a VQ‑style method. While both approaches share the idea of using prototype‑based representations, there are fundamental differences in their motivation, mechanism, and application, as detailed below:
>
> 1. **Purpose and Conceptual Distinction**
>     VQ is primarily designed to _quantize_ continuous representations into discrete codes through a hard nearest‑neighbor assignment to a fixed codebook, often with commitment losses to stabilize training. This makes VQ well‑suited for bridging continuous and discrete representations (e.g., in VQ‑VAE). In contrast, our method does **not** quantize representations. Instead of selecting a single prototype, we **retain the entire soft distribution** over prototypes as the target. This preserves uncertainty and provides richer supervision via _soft targets_, which is particularly beneficial in masked atom modeling, where multiple reconstructions can be semantically plausible.
>
> 2. **Soft Labels vs. Hard Assignments**
>     VQ’s hard assignments inherently discard uncertainty and can introduce gradient mismatch. Our approach deliberately avoids hard assignments by using a **temperature‑controlled softmax** over cosine similarities to prototypes.
>
> 3. **Task‑Specific Motivation**
>     Masked atom modeling often involves chemically similar atoms as potential reconstructions. Hard targets (as in VQ or standard classification) ignore this nuance. Our soft label design explicitly leverages **temperature‑scaled similarity** to learnable prototypes, generating probabilistic targets that better capture chemical variability.
>
> 4. **Prototype Utilization and Regularization**
>     To further enhance learning, we introduce **Mean Entropy Maximization (ME‑MAX)** regularization, which encourages balanced prototype utilization and prevents collapse. This regularization is not typically considered in VQ approaches.
>
> 5. **Empirical Effectiveness**
>     As shown in our experiments, the proposed soft label method consistently outperforms hard label (e.g., AttrMask) and VQ‑style (e.g., Mole-Bert) baselines. We attribute this to its ability to encode uncertainty and chemical similarity through soft supervision.
>
>
> **Summary:**
> While both methods involve prototypes, our soft label approach fundamentally differs from VQ by **eschewing quantization**, embracing **soft distributions** as supervision, and explicitly modeling uncertainty. These differences yield richer training signals, more robust prototype usage, and ultimately better generalization, particularly in the context of molecular masked modeling.
>
>
> ---
>
>
>
>
> > **Q4.** _I think the $\alpha H\left(s^{\bar{y}} p\right)$   in equation 17 should be $\alpha H\left(\bar{s}^{\bar{p}}\right)$ but I am not sure. The formula seems to have a typo, though._
>
>
> We sincerely thank the reviewer for carefully checking our equations. You are correct that there is a typo in Equation 17. The correct formulation should be:
>
> $$
> \alpha H\left(\bar{s}^{\bar{p}}\right)
> $$
>
> instead of
>
> $$
> \alpha H\left(s^{\bar{y}} p\right)
> $$
>
> ___
>
> We hope these explanations and corrections satisfactorily address your concerns. Thank you again for your thoughtful review.

---

### Official Review · Reviewer_oeWH · 2025-07-02

**Clarity:** 2
**Significance:** 2
**Originality:** 2
**Rating:** 3
**Confidence:** 4

**Summary:**

This paper introduces a new method of dynamic masking strategy to enhance molecular masked graph autoencoder. The authors first show that a static mask ratio and random graph corruption are suboptimal in molecular graph pretraining with masking-reconstruction loss. To address this issue, this paper utilize Graph Information Bottleneck to calculate an importance socre of each atom, based on which a masking probability is determined, so that a adaptive masking strategy can be applied to each molecule. Experiments are conducted on a dataset sourced from the 252 ZINC15 database and the proposed strategy are integrated into three different Masked Graph Autoencoders for molecular representation learning. Experimental results show that the proposed method enhance the performance of the three Masked Graph Autoencoders and the integrated models outperform existing a series of molecular pretraining methods.

**Questions:**

1. Please respond to the weakness points.
2. Does the proposed method bring high time cost to pretraining?

**Ethical Concerns:**

["NO or VERY MINOR ethics concerns only"]

**Final Justification:**

My major concern was the weak baselines in the experiments. In the response, more recent methods are added in comparison and with these new baselines, the performance gains looks even less significant.  Therefore, I keep my original score.

**Limitations:**

Yes

**Quality:**

2

**Strengths And Weaknesses:**

Strength
1.It is valuable to show that dynamic masking based on the importance score of each atom is superior than random masking with a fixed masking ratio in masked graph autoencoder pretraining.
2.The paper is generally well written and easy to follow and the motivation is reasonable.
3.The strategy of using information bottleneck to calculate an importance score of each atom is feasible. The contribution of using information bottleneck in an unsupervised scenario is valuable.

Weakness
1.The major weakness is that the baseline methods are weak and outdated. The proposed method is compared to quite a lot self-supervised graph pretraining models, but the results are from MoleBert [41], which is published in 2023. After the publication of [41], self-supervised pretraining for molecular representation learning has advanced rapidly and quite a lot new methods have been proposed. A search of papers citing [41] can find some of them. Here I list several papers that the authors may consider.

[1] Hongxin Xiang, et al. An Image-enhanced Molecular Graph Representation Learning Framework. IJCAI 2024
[2] Liang Wang, et al. MOLSPECTRA: PRE-TRAINING 3D MOLECULAR REPRESENTATION  WITH MULTI-MODAL ENERGY SPECTRA. ICLR 2025.
[3] Yue Wan, et al. Multi-channel learning for integrating structural hierarchies into contextdependent  molecular representation. Nature Communications 2025.

2.Relevant to the first point, this paper lacks a review of self-supervised molecular pre-training techniques. A comprehensive review of this field can place the current research in the correct context.

3.The proposed method is only evaluated on one dataset.

4.The is not comparison to conventional masking strategies with proper hyper-parameter selection.

---

> ### Author Rebuttal · Authors · 2025-07-30
>
> Thank you very much for your valuable feedback and suggestions. Below, we address each point in turn.
>
> ---
>
> > **Q1.** _Weakness 1. The major weakness is that the baseline methods are weak and outdated. The proposed method is compared to quite a lot self-supervised graph pretraining models, but the results are from MoleBert [41], which is published in 2023. After the publication of [41], self-supervised pretraining for molecular representation learning has advanced rapidly and quite a lot new methods have been proposed. A search of papers citing [41] can find some of them. Here I list several papers that the authors may consider._
> > [1] Hongxin Xiang, et al. IJCAI 2024
> > [2] Liang Wang, et al. ICLR 2025
> > [3] Yue Wan, et al. Nature Communications 2025
>
> Thank you for pointing this out and for listing several recent works. We agree that the baseline set can be further strengthened and have updated the experiments accordingly.
>
> 1. **Expanded baseline set.** In addition to the MGAE‑style methods originally reported (AttrMask [2020], GraphMVP [2021], Mole‑BERT [2023], and SimSGT [2024]), we have added the recent models you suggested (IEM‑GraphMVP, IEM‑Mole‑BERT, StructMAE‑P/L). The updated table below reports their numbers on the same scaffold‑split evaluation protocol as used in our paper:
>
> |Method|Tox21|ToxCast|Sider|ClinTox|MUV|HIV|BBBP|BACE|**Avg.**|
> |---|---|---|---|---|---|---|---|---|---|
> |IEM‑GraphMVP|75.6|64.8|62.0|79.2|77.0|78.2|71.4|81.9|**73.8**|
> |IEM‑Mole‑BERT|77.8|65.6|65.3|79.7|72.2|78.8|68.1|83.0|**73.8**|
> |StructMAE‑P|75.8|64.5|62.0|86.0|77.7|77.4|84.3|72.6|**75.0**|
> |StructMAE‑L|75.3|64.0|61.3|87.9|78.0|78.3|72.5|83.2|**75.1**|
> |**SimSGT‑DyCC‑GraphTrans (Ours)**|76.6|66.3|62.0|83.6|80.3|77.8|72.2|84.9|**75.5**|
>
> The strongest average result among the new baselines is 75.1 (StructMAE‑L), which is still lower than our **SimSGT‑DyCC‑GraphTrans** (75.5) (see Table 6).
>
> 2. **Scope clarification.** Our primary goal is not only to chase SOTA, but to diagnose limitations of existing molecular masked pre‑training paradigms (e.g., fixed masking ratio, rigid reconstruction targets) and propose a general‑purpose solution (**DyCC**) that can be plugged into a variety of MGAEs.
>
> 3. **References and discussion.** We will add the recent works you suggested (IEM‑GraphMVP, IEM‑Mole‑BERT, StructMAE) and others published after 2023 into the related‑work section and discuss how DyCC differs conceptually.
>
>
> In summary, the updated results show that our approach remains competitive or superior to recent methods, while being a **model‑agnostic plug‑in** rather than a full‑stack redesign.
>
> ---
>
> > **Q2.** _This paper lacks a review of self-supervised molecular pre-training techniques. A comprehensive review of this field can place the current research in the correct context._
>
> Thank you for this constructive suggestion. We agree that a more comprehensive overview will improve the clarity and positioning of our contribution.
>
> In **Section 2**, we currently review two main paradigms of self‑supervised molecular graph pretraining:
>
> - **Graph contrastive learning**, e.g., GraphCL, JOAO, GraphLOG, RGCL, SimGRACE
>
> - **Graph generative (masked) learning**, e.g., AttrMask, GraphMAE, Mole‑BERT, SimSGT
>
>
> These representative methods are directly relevant to our proposed DyCC framework. In addition, **Appendix C.3** provides detailed descriptions of all baselines used in Table 1.
>
> That said, we acknowledge that the field has been evolving rapidly, and several recent works were not explicitly covered. In the revised manuscript, we will expand **Section 2** into a broader taxonomy of self‑supervised molecular pretraining, including **multi‑view/multi‑modal methods**, **contrastive learning**, **3D‑aware and SE(3)‑equivariant models**, and **generative objectives**.
>
>
> ---
>
> > **Q3.** _The proposed method is only evaluated on one dataset._
>
> We respectfully clarify that our method is evaluated on **multiple datasets and task types**, not just a single dataset:
>
> - In **Section 4 (Table 1)**, we report results on **eight classification tasks** from MoleculeNet.
>
> - In **Appendix Table 5**, we further evaluate on **four molecular property regression tasks** and **two drug–target affinity (DTA) regression tasks**.
>
> - In **Appendix Table 7**, we also report results on **quantum chemistry property prediction** tasks.
>
>
> These evaluations span different domains (classification, regression, and quantum property prediction), demonstrating the **robustness and generality** of the proposed method.
>
> ---
>
> > **Q4.** _There is no comparison to conventional masking strategies with proper hyper-parameter selection._
>
> We would like to clarify that for **AttrMask**, **Mole‑BERT**, and **SimSGT**, the masking strategy is primarily controlled by the **masking ratio**. The authors of these methods have already conducted extensive hyper‑parameter studies in their original papers, and we follow their **default settings** to ensure fair and consistent comparison.
>
> For example, Mole‑BERT (MAM) reports results across multiple masking ratios in Table 7 of the original paper (reproduced below), showing that 0.20 is near optimal:
>
> |Masking ratio|0.10|0.15|0.20|0.25|0.30|
> |---|---|---|---|---|---|
> |MAM|71.64|72.16|**72.21**|71.51|71.36|
>
> Our **AttrMask‑DyCC**, **Mole‑BERT‑DyCC**, and **SimSGT‑DyCC** experiments keep the same experimental setup and hyper‑parameter configurations as the corresponding baselines. We did not conduct additional hyper‑parameter searches to avoid introducing bias and to ensure a fair comparison.
>
> ---
>
> > **Q5.** _Does the proposed method bring high time cost to pretraining?_
>
> We appreciate the reviewer’s concern.
>
> **Theoretical analysis:** Our method introduces two lightweight modules—**GIBMS** and **SLG**.
>
> - GIBMS is trained **once and reused** across all runs, so its cost is amortized.
>
> - SLG only adds two matrix multiplications (Eq. 14 and Eq. 15: $\mathbf{H}^y \cdot \mathbf{Q}$ and $\mathbf{H}^p \cdot \mathbf{Q}$), which introduce **negligible computational overhead**.
>
>
> **Empirical results:** All experiments were conducted on an NVIDIA DGX A100 server.
>
> - GIBMS training takes **683 minutes** (one‑time).
>
> - Adding SLG to AttrMask increases pretraining time from **512 min to 561 min**, a modest 8.7% increase.
>
>
> Thus, the proposed method achieves performance gains with only a **small and manageable time cost**.
>
> ___
>
> We hope these explanations and corrections satisfactorily address your concerns. Thank you again for your thoughtful review.

---

> > ### Author Response · Authors · 2025-08-06
> >
> > Dear Reviewer,
> >
> > Thank you for your thorough review and helpful feedback. We did our best to address your concerns in the rebuttal and would appreciate your guidance on whether our responses resolved them. We are glad to provide any further details or results if helpful.
> >
> > With appreciation,
> >
> > Authors

---

> > ### Comment · Reviewer_oeWH · 2025-08-07
> > **further comments**
> >
> > I appriciate the detailed response and most of my concerns are resolved. However, with new baselines added, the performance gain of the proposed method becomes more insignificant. I understand the claim that "Our primary goal is not only to chase SOTA, but to diagnose limitations of existing molecular masked pre‑training paradigms", but if essential performance gain can only be achieved on the base of weak baselines, the diagnose of limitation seems less valuable. Therefore, I tend to keep my original score.

---

> > > ### Author Response · Authors · 2025-08-07
> > >
> > > **Dear Reviewer,**
> > >
> > > Thank you very much for your thoughtful observations. We understand that your concern may arise from the relatively limited performance gains observed after incorporating StructMAE into our framework. We sincerely appreciate the opportunity to clarify this point in more detail.
> > >
> > > 1. **On Pathways to Achieve Higher Performance**
> > >     We agree that further performance improvements can be achieved through various strategies, such as incorporating **external domain knowledge** (e.g., molecular fingerprints [1]), leveraging **multimodal information** (e.g., combining 2D and 3D molecular representations [2]), or training on **larger-scale datasets** [3]. However, these approaches go beyond the scope of our current study. Our work is focused on advancing the **core MGAE pretraining paradigm**, and we believe that maintaining this focus ensures both conceptual clarity and methodological rigor.
> > >
> > > 2. **On Interpretability and Reusability**
> > >     Our method also brings added **interpretability and reusability**. While StructMAE and our proposed GIBMS module share a similar underlying intuition—that different atoms should be treated with different importance during masking—GIBMS is grounded in **Information Bottleneck (IB) theory**, offering a principled framework to evaluate and exploit atom-level informativeness. In contrast, StructMAE employs a learnable sampling strategy without a theoretical basis. Moreover, **GIBMS is designed as a reusable, modular component** that can be flexibly integrated into a variety of architectures, while StructMAE is more tightly bound to a specific design.
> > >
> > > 3. **Beyond DyCC: The Challenge of Benchmark Saturation**
> > >     We would also like to emphasize that the **challenge is not limited to DyCC**, but reflects a broader issue of **benchmark saturation** in the field. As noted in recent work, many MoleculeNet tasks have seen top-performing models (e.g., Mole-BERT, SimSGT) approach the upper limits of achievable performance. In such scenarios, even **small improvements (e.g., 1–2% ROC-AUC)** can be **practically significant**, especially for downstream applications like **virtual screening** or **toxicity prediction**, where marginal gains can lead to improved hit rates and real-world impact. For further insights into the limitations of current small-molecule benchmarks, we refer to Kretschmer _et al._ [4], which offers a detailed analysis of **coverage bias** and other structural constraints in widely used datasets.
> > >
> > >
> > > We hope these clarifications help contextualize both the scope and the contributions of our work. Please don’t hesitate to let us know if there are further points you would like us to elaborate on—we would be more than happy to continue the discussion.
> > >
> > > Warm regards,
> > > The Authors
> > >
> > > ---
> > >
> > > **References:**
> > > [1] Li, Han, et al. _A knowledge-guided pre-training framework for improving molecular representation learning_. _Nature Communications_, 14.1 (2023): 7568.
> > > [2] Zhu, Jinhua, et al. _Unified 2D and 3D pre-training of molecular representations_. _Proceedings of the 28th ACM SIGKDD_, 2022.
> > > [3] Ji, Xiaohong, et al. _Uni-Mol2: Exploring molecular pretraining model at scale_. _arXiv preprint arXiv:2406.14969_, 2024.
> > > [4] Kretschmer, Fleming, et al. _Coverage bias in small molecule machine learning_. _Nature Communications_, 16.1 (2025): 554.

---

### Note · Authors · 2025-08-12

**Dear (Senior) ACs and Reviewers,**

We would like to express our heartfelt thanks for the time, care, and expertise you have devoted to reviewing our manuscript. We are truly grateful for the constructive feedback and thoughtful suggestions, which have been invaluable in helping us refine and strengthen our work.

During the rebuttal phase, the reviewers kindly raised several important points, such as adding suitable baselines, comparing to conventional masking strategies with proper hyperparameter selection, clarifying the differences between the SLG module and vector quantization (VQ), examining the stability of atom importance scores across runs, analyzing computational overhead, assessing the significance of DyCC’s performance gains, evaluating the effectiveness of GIBMS in subgraph identification, and providing interpretability analysis of the IBF module.

Among these, we understand that the significance of DyCC’s improvements was of particular interest. Briefly:

1. **Effective beyond weaker baselines.** DyCC proves beneficial not only for relatively weaker baselines (e.g., MGAEs) but also for strong, recent models such as SimSGT (2024) and StructMAE (2025), surpassing both when integrated into SimSGT.

2. **Meaningful and significant gains.** DyCC yields improvements of ~1.5%–4.3% on downstream tasks. Repeated experiments confirm these gains are statistically significant (p-value < 0.001). In real-world scenarios such as virtual screening, even 1–2% ROC-AUC gains can lead to **notably higher hit rates** and reduced experimental costs.

3. **Dataset limitations, not method limitations.** Current benchmarks show performance saturation, which constrains measurable improvements. This reflects dataset limitations rather than any inherent limitation of DyCC.

4. **Beyond chasing SOTA.** Our goal is to address general challenges in MGAEs-based molecular pretraining. While incorporating domain knowledge, multimodal inputs, or larger datasets may further enhance performance, these directions are beyond the current scope.


Once again, we sincerely appreciate your thoughtful evaluation, the generous feedback, and the opportunity to clarify and improve our work.

---

### Decision · Program_Chairs · 2025-09-17

**Decision:**

Accept (poster)

**Comment:**

The paper proposes Dynamic and Chemical Constraints (DyCC) to enhance masked graph autoencoders for molecular representation learning, introducing a graph information bottleneck-based masking strategy (GIBMS) and a soft label generator (SLG) to address limitations in fixed masking ratios and reconstruction objectives. Reviewers praised the method's theoretical grounding, model-agnostic design, and consistent performance improvements across multiple baselines and tasks, though one noted modest gains on saturated benchmarks and initially leaned toward rejection before discussions. Strengths include the extension of information bottleneck theory to unsupervised settings, thorough empirical validation on MoleculeNet datasets, and interpretability of atom importance scores, with rebuttals effectively addressing concerns about baselines, computational overhead, and statistical significance. Weaknesses, such as reliance on older baselines and minor presentation issues, were mitigated through additional experiments and clarifications in the rebuttal phase. Overall, the contributions are solid and impactful for molecular pretraining, warranting acceptance at NeurIPS 2025.